# Rainbow Delay Compensation: A Multi-Agent Reinforcement Learning Framework for Mitigating Delayed Observation

**Songchen Fu**[1,2,*]**, Siang Chen**[3,*]**, Shaojing Zhao**[1,2]**, Letian Bai**[1,2]**, Hong Liang**[1,2]**,
Ta Li**[1,2,✉]**, Yonghong Yan**[1,2]

[1]Laboratory of Speech and Intelligent Information Processing, Institute of Acoustics, CAS
[2]University of Chinese Academy of Sciences
[3]Department of Electronic Engineering, Tsinghua University
`fusongchen@hccl.ioa.ac.cn, lita@hccl.ioa.ac.cn`

## Abstract

In real-world multi-agent systems (MASs), observation delays are ubiquitous, preventing agents from making decisions based on the environment's true state. An individual agent's local observation typically comprises multiple components from other agents or dynamic entities within the environment. These discrete observation components with varying delay characteristics pose significant challenges for multi-agent reinforcement learning (MARL). In this paper, we first formulate the decentralized stochastic individual delay partially observable Markov decision process (DSID-POMDP) by extending the standard Dec-POMDP. We then propose the Rainbow Delay Compensation (RDC), a MARL training framework for addressing stochastic individual delays, along with recommended implementations for its constituent modules. We implement the DSID-POMDP's observation generation pattern using standard MARL benchmarks, including MPE and SMAC. Experiments demonstrate that baseline MARL methods suffer severe performance degradation under fixed and unfixed delays. The RDC-enhanced approach mitigates this issue, remarkably achieving ideal delay-free performance in certain delay scenarios while maintaining generalizability. Our work provides a novel perspective on multi-agent delayed observation problems and offers an effective solution framework. The source code is available at `https://github.com/linkjoker1006/RDC-pymarl`.

## 1 Introduction

Multi-Agent reinforcement learning (MARL) has been widely applied in various domains such as multiplayer games [28, 19], robot control [13, 14], agent communication [5, 40], and quantitative trading [37]. However, beyond inherent challenges in MARL, including non-stationarity, partial observability, credit assignment, and the curse of dimensionality [38, 12], the observation delay problem has often been overlooked. From signal transmission in biological systems [11] to communication in large-scale swarms [29], delay issues are ubiquitous in real-world scenarios and typically have detrimental effects on most systems. Due to the coupled influences of the environment, allied agents, and other agents (opponents or targets), multi-agent systems (MASs) exhibit more prevalent and complex observation delay situations than single-agent systems.

Early studies on system delay problems were primarily rooted in control theory [1, 23], where solutions relied heavily on fixed transition models—an assumption often violated in complex MASs [25].

---

*These authors contributed equally to this work.

The introduction of augmented state spaces [2, 32] marked a pivotal shift, enabling reinforcement learning methods to handle deterministic delays through model-based state estimation [9, 7]. While these approaches advanced single-agent systems, their extension to multi-agent settings remained superficial, typically limited to fixed delay scenarios [7, 33, 21]. Recent progress in delayed-observation Markov decision processes (DOMDPs) [34] formalized stochastic delay modeling, yet existing work concentrates overwhelmingly on single-agent domains. Multi-agent solutions [36, 35] established theoretical foundations and algorithmic innovations at the levels of communication and feedback. Yet, a critical gap remains: the fundamental challenge of stochastic partial observability in MASs remains unresolved. This oversight is particularly significant given the inherent asynchrony and network-induced uncertainties in real-world multi-agent applications.

In MARL, the evolving policies of other agents render the learning environment inherently non-stationary. Stochastic observation delays exacerbate this challenge, as agents must not only cope with non-stationarity but also account for the uncertainty introduced by varying delays. These delays also exacerbate the credit assignment problem: misaligned observations and rewards make it more challenging to associate actions with outcomes, thereby hindering effective policy optimization. Fundamentally, unfixed delays violate the Markov assumption by providing agents with inaccurate perceptions of the current state. Moreover, unfixed delays have a greater impact than fixed ones. With fixed delays, agents can adapt and implicitly predict others' observations over time, forming a kind of cognitive inertia. However, under stochastic delays, such predictions become unreliable, making it harder for actor networks to compensate. These facts all indicate that the problem of unfixed delayed observations in MARL is in urgent need of being addressed.

Previous studies have primarily focused on delay issues in single-agent systems, whereas research on MARL mainly consists of simple extensions of single-agent delay theories. However, delays in multi-agent environments are not constant, and different observation components may experience varying delays for each agent. Our work provides a deeper understanding of delayed observation in MASs and proposes a general algorithmic framework to address the associated challenges. This means practitioners can tailor the framework by selecting and integrating suitable algorithms for different scenarios, achieving optimal performance. The main contributions of this paper are as follows:

- We define the decentralized stochastic individual delay POMDP (DSID-POMDP), providing a universal mathematical model for MASs with delayed observation.

- We propose the Rainbow Delay Compensation (RDC) training framework, which mitigates the impact of delayed observation by reconstructing delay-free observation, utilizing curriculum learning, and leveraging knowledge distillation.

- Based on DSID-POMDP, we innovatively introduce two compensator operation modes (Echo and Flash) and implement two models based on Transformer and Gated Recurrent Unit (GRU) networks.

- We integrate two classic MARL algorithms, VDN [30] and QMIX [27], into the RDC framework. The proposed method is tested on two common MARL benchmarks, demonstrating significant performance improvements under fixed and unfixed delay conditions, approaching near delay-free performance levels.

## 2 Related Works

Recent studies have explored Deep Reinforcement Learning (DRL) approaches to address delay issues in single-agent systems. Walsh et al. [32] introduced the constant delay Markov decision process (CDMDP), which extends action sequences to incorporate fixed delays in observation and reward. However, the resulting state space expansion suffers from exponential growth, limiting the feasibility of pure state-based solutions. To overcome this, they proposed Model-Based Simulation (MBS) for discrete environments and Model Parameter Approximation (MPA) for continuous environments, pioneering model-based methods for delay-free state estimation. Firoiu et al. [9] employed environment prediction models to reduce performance degradation from fixed action delays in gaming scenarios. Bouteiller et al. [4] developed a partial trajectory resampling method to resolve credit assignment challenges in stochastic delay environments. Liotet et al. [20] implemented imitation learning to align delayed agents with expert action distributions, but only for fixed delays. Wang et al.

[34] combined state augmentation and state prediction with delay-reconciled training for separate actor-critic optimization, demonstrating significant improvements in stochastic delay scenarios.

Significant progress has also been made in addressing the challenges of delay in MARL. Chen et al. [6] developed the Delay-Aware Multi-Agent Reinforcement Learning (DAMARL) framework, which mitigates fixed observation delays through centralized training with auxiliary information. Subsequent work by Yuan et al. [36] introduced TimeNet to dynamically optimize agents' waiting time for delayed communications, thereby enhancing collaborative efficiency. For reward delay scenarios, Zhang et al. [39] proposed Delay-Adaptive Multi-Agent V-Learning (DAMAVL) with proven convergence under finite and infinite delay conditions. Practical applications have shown promise, as demonstrated by Liu et al. [21]'s successful implementation of DAMARL in cooperative adaptive cruise control (CACC) systems. While Wang et al. [33] advanced the field by predicting action effects through state prediction. Still, current approaches remain limited to fixed delay scenarios and typically overlook the asynchronicity of delayed observations. It is worth noting that research on robust MARL [15] also aims to address the challenges of non-ideal observations. Unlike the observation lag caused by delays, this line of work focuses more on inaccuracies in observations resulting from system errors or adversarial attacks.

# 3 Preliminaries

## 3.1 Decentralized Partially Observable Markov Decision Process

A decentralized partially observable Markov decision process (Dec-POMDP) is a model designed for coordination and decision-making in MASs. In this framework, the POMDP introduces an "observation" variable, enabling decision-makers to observe only a portion of the system state at each time step, building upon the original MDP. Dec-POMDP extends this concept to multi-agent scenarios by incorporating additional joint variables. It can be formally defined as a 7-tuple $(\mathcal{I}, \mathcal{S}, \mathcal{A}, \mathcal{Z}, \mathcal{P}, \mathcal{R}, \mathcal{O})$, where $\mathcal{I}$ represents a finite set of agents. $\mathcal{S}$ denotes the set of state systems encompassing all possible environment states. $\mathcal{A}$ and $\mathcal{Z}$ are the action and observation spaces for each agent, respectively, with $\mathcal{A} = \mathbf{A}_i$ and $\mathcal{Z} = \mathbf{Z}_i$. $\mathcal{P}, \mathcal{R},$ and $\mathcal{O}$ are defined as $\mathcal{P}(\mathbf{s}'|\mathbf{s}, \mathbf{a}), \mathcal{R}(\mathbf{r}'|\mathbf{s}, \mathbf{a}),$ and $\mathcal{O}(\mathbf{o}|\mathbf{s})$, representing the probabilities of transitioning to state $\mathbf{s}'$, receiving reward $\mathbf{r}'$, and obtaining observation $\mathbf{o}$ after executing the joint action $\mathbf{a}$ in state $\mathbf{s}$. This formulation captures the decentralized nature of decision-making in MASs, where agents must coordinate their actions based on partial observation of the environment.

## 3.2 Decentralized Stochastic Individual Delay-Partially Observable Markov Decision Process

While delayed observation MDPs represent a special case of POMDPs, we maintain a clear distinction between delayed and partial observation in this paper. In typical multi-agent environments, each agent's observation comprises three components: (1) self-state information (typically with minimal delays due to internal transmission), (2) other agents' states (obtained through perception or communication), and (3) environmental states. Since delays in observing other agents and environmental states follow similar principles, we collectively term these observation sources as "entities" for clarity. The observation delays from other entities are typically positively correlated with their relative distances. Inspired by this phenomenon, we can model the possible delay values of different entities in $agent_i$'s observation as multiple user-defined probability distributions, not just those

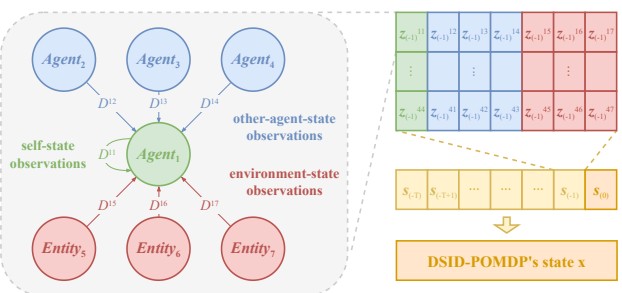

Figure 1: A simple example of extended state and delayed observation in DSID-POMDP. The left side describes the components of $agent_1$'s observation and annotates their delay value distributions. The matrix in the upper right corner shows the contents of $s_{(-1)}$ in the extended state.

related to distance. Therefore, we propose the decentralized stochastic individual delay POMDP (DSID-POMDP).

**Definition 1.** A DSID-POMDP $= (\mathcal{I}, \mathcal{X}, \mathcal{A}, \mathcal{Z}, \mathcal{D}, \mathcal{P}_{\mathcal{D}}, \mathcal{R}_{\mathcal{D}}, \mathcal{O}_{\mathcal{D}})$ augments a Dec-POMDP $= (\mathcal{I}, \mathcal{S}, \mathcal{A}, \mathcal{Z}, \mathcal{P}, \mathcal{R}, \mathcal{O})$, such that

1. $\mathcal{I}_{\mathcal{D}} = \mathcal{I} \cup \mathcal{J}$ where $\mathcal{J}$ is the set of environment entities,
2. $\mathcal{X} = \mathcal{S}^{T+1}$ where $T$ denotes the maximum possible delay value,
3. $\mathcal{A}_{\mathcal{D}} = \mathcal{A}$,
4. $\mathcal{Z}_{\mathcal{D}} = \mathcal{Z}$,
5. $\mathcal{D}^{ij} = \mathcal{D}(agent_i, entity_j | \mathbf{x}), i \in \mathcal{I}, j \in \mathcal{I}_{\mathcal{D}}$,
6. $\mathcal{P}_{\mathcal{D}}(\mathbf{x}'|\mathbf{x}, \mathbf{a}) = \mathcal{P}(s'|s, \mathbf{a}) \prod_{t=1}^{T} \delta(s'_{(-t)} - s_{(-t+1)}), s'_{(-t)} \in \mathbf{x}', s_{(-t+1)} \in \mathbf{x}$,
7. $\mathcal{R}_{\mathcal{D}}(\mathbf{x}, \mathbf{a}) = \mathcal{R}(s, \mathbf{a})$,
8. $\mathcal{O}_{\mathcal{D}}(\mathbf{z}|\mathbf{x}) = \prod_{i \in \mathcal{I}} \prod_{j \in \mathcal{I}_{\mathcal{D}}} \sum_{t=0}^{T} p(d^{ij} = t) \delta(z^{ij} - s^{ij}_{(-t)}) \mathcal{O}(o^{ij} = z^{ij} | s_{(-t)})$.

The DSID-POMDP is established under the condition that the information exposed by each entity in the system only contains information from itself. The new element, individual delay distribution $D^{ij}$, represents the distribution of delay values for $entity_j$ in the observation of $agent_i$. A natural and intuitive constraint is that the delay value $d_t^{ij}$ must satisfy the condition: $d_t^{ij} < min(d_{t-1}^{ij} + 1, T)$. To incorporate delayed states, the state of the DSID-POMDP is extended to include the delay-free state and the previous $T$ states, that is, $\mathbf{x} = \{s_{(-T)}, s_{(-T+1)}, ..., s_{(-1)}, s\}$. To maintain consistency in notation, we also represent the delay-free state $s$ as $s_{(0)}$. In the observation function, $p(d^{ij} = t)$ represents the probability that the delay value $d^{ij}$ equals $t$, and $s^{ij}_{(-t)}$ represents the state information of $entity_j$ from the perspective of $agent_i$ in state $s_{(-t)}$. Figure 1 provides a more intuitive explanation of individual delay distributions and the observation function.

### 3.3 Classic MARL Algorithms

The operation of the RDC framework relies on baseline algorithms. The classification of MARL algorithms follows a similar pattern to DRL, where they can be categorized into value-based and policy-based methods according to their underlying principles. After comprehensively considering algorithm performance and characteristics, we employ two value-based algorithms in our framework, VDN [30] and QMIX [27], which demonstrate excellent performance in discrete action space tasks. VDN extends DQN [24] in a straightforward manner by decomposing the team value function using a linear factorization approach. During training, VDN samples batches from the replay buffer and updates parameters by minimizing the TD error, with its loss function defined as:

$$\mathcal{L}_{rl}(\theta) = \hat{\mathbb{E}}_b[(r + \gamma \max_{\mathbf{u}'} Q_{total}(s', \mathbf{u}'; \theta^-) - Q_{total}(s, \mathbf{u}; \theta))^2], \tag{1}$$

where $\theta^-$ represents the parameters of the target network. The target network periodically copies (hard update) or gradually weights (soft update) the parameters $\theta$ of the evaluation network. $\hat{\mathbb{E}}_b$ denotes the expectation over a finite batch of samples.

Building upon VDN, QMIX uses a mixing network to aggregate the local Q-values of individual agents into a centralized global Q-value, while satisfying the monotonicity constraint: $\frac{\partial Q_{total}}{\partial Q_i} \geq 0, \forall i$. This ensures consistency between the global and local Q-values. This modification endows QMIX with a parameterized critic, whereas VDN relies solely on a simple summation operation. This distinction serves to validate RDC's adaptability to different RL algorithm frameworks.

## 4 Methods

In this section, we present the architectural details of RDC. As illustrated in Figure 2, the framework extends conventional MARL algorithms by incorporating four key components: 1) a compensator module, 2) a delay-reconciled critic, 3) the curriculum learning for actors, and 4) the policy knowledge distillation. The arrows in different colors and styles represent the data flows in various stages. Blue arrows belong to the teacher model, and red arrows belong to the student model. Solid arrows denote the training phase, while dashed arrows represent the inference phase. The data flows involved in the inference phase can be used during the training phase. While we describe several viable

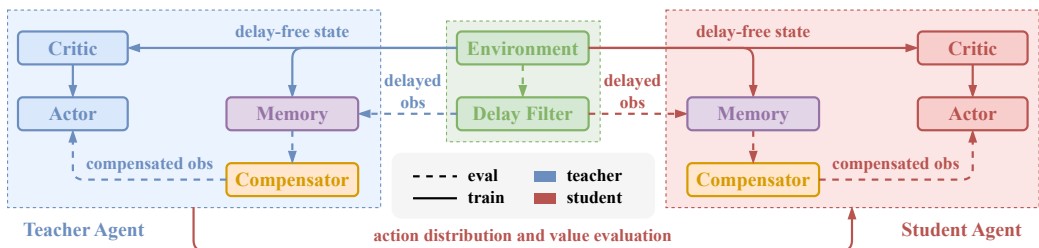

Figure 2: The internal structure of the DA-MARL framework.

implementations and corresponding algorithms, these do not represent an exhaustive enumeration of possibilities within this framework. The framework maintains broad compatibility - most mainstream actor-critic algorithms can seamlessly adapt to RDC. At the same time, other MARL approaches can typically be accommodated by selectively removing specific framework components. The compensator may employ any sequence prediction-capable architecture, and both the curriculum learning and knowledge distillation mechanisms can be customized by researchers based on particular policy networks and task requirements.

## 4.1 Observation Delay Occurrence and Compensation Process in Multi-agent System

Due to the mutually independent action delays among agents in MASs, and each agent's actions can influence the global state, the delay equivalence theorem [34] cannot hold. This paper focuses exclusively on delayed observation and assumes that agents only transmit their own information externally—an assumption consistent with most multi-agent environment configurations in the research community. Therefore, from the viewpoint of a single agent, information updates from different entities are relatively independent of each other. This principle leads to the phenomenon where other

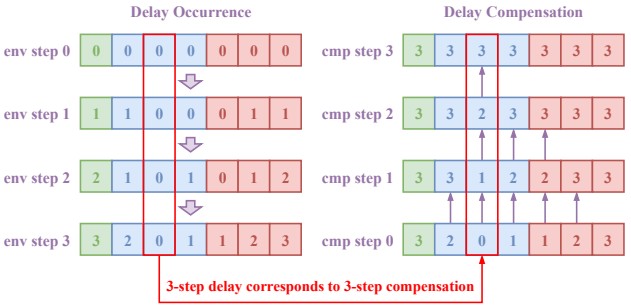

Figure 3: A simplified example illustrating MASs' delay occurrence and compensation process.

parts of the observation do not belong to the same time step. Figure 3 illustrates a simple example of delayed observation. The numbers indicate the time step from which each part of the agent's current observation originates — smaller numbers correspond to older information. The left side demonstrates the information updates in the observations of $agent_i$ over system clock steps 0 to 3. In contrast, the right side shows the corresponding compensation process of observations without delay. The observation from the second agent at step 3 still contains information from step 0, requiring the most compensation steps for this portion.

## 4.2 Delay Compensator

Based on the occurrence process of observation delay, we intuitively design two modes of the delay compensator—*Flash* and *Echo*—to reconstruct delay-free observation. *Flash* performs a simple compensation using available information and directly outputs the reconstructed result. It implicitly accounts for the issue of varying delays within observation through its internal model design. *Echo*, an autoregressive model, incrementally outputs the next-step information based on known data at each compensation step. Under the masking layer's control, the $T$-th output of *Echo* yields the final reconstructed observation. Figure 5 illustrates the workflows of the two compensators in parallel. Theoretically speaking, *Flash* offers faster reconstruction with lower resource consumption, making it suitable for scenarios with slight delay variations and high requirements for decision-making speed. *Echo*'s operation mode fully complies with the ideal delay compensation process shown in Figure 3, and can adapt to variable delay values and unknown delay patterns. We believe that as

agent policies iterate and update, the data distribution will change, potentially causing significant performance fluctuations in the compensators. Thus, both compensators are trained synchronously with reinforcement learning, meaning the RDC framework maintains a complete online training process.

Sufficient and effective input data enables the compensator to achieve better performance. Based on the definition of DSID-POMDP, we naturally extend the current observation into an observation sequence that includes history observations. Furthermore, drawing on existing augmentation methods [6, 7, 34, 33], we empirically incorporate action sequences from the past $T$ timesteps. Unlike previous approaches, the delay steps in this paper are represented as a vector whose length corresponds to the number of other entities included in the observation. We implement the two types of compensators using GRU [8] and Transformer [31] networks, which excel at processing sequential data. Figure 4 illustrates the extended observation input forms when different model structures are employed. Fixed inputs refer to the pre-concatenated sequence data fed into the model during the first input step. In contrast, sequential inputs represent the incrementally augmented input information during the model's autoregressive process. For *Echo*, we convert the delay value vector into a binary (0 or 1) vector to enhance consistency between history and autoregressive inputs.

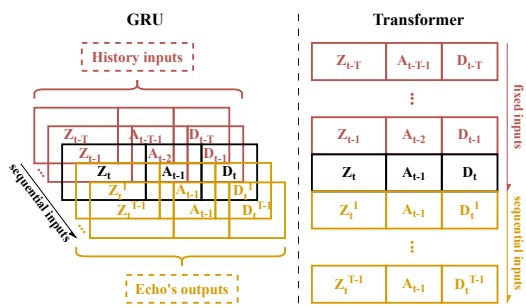

Figure 4: Inputs of compensators with different modes and networks. The figure only illustrates the input formation of a single $agent_i$ at timestep $t$, and for clarity, we omit the subscript $i$. The subscript denotes the timestep in the environment iteration, while the superscript indicates both the $k$-th output of *Echo* and the $(k{+}1)$-th input. Inputs of *Flash* exclude the yellow-highlighted components.

Additionally, we design a dual-head residual compensator to handle different data types in observations. We employ the cross-entropy loss function for the classification task, while utilizing the mean squared error for the regression task. The residual output can further improve the reconstruction accuracy of the compensator. The loss functions for the two compensators are as follows:

$$\mathcal{L}_{\text{flash}}(\phi) = \mathcal{L}_{\text{CE}}(\mathbf{I}^T, \mathbf{I}(Z^{T\_GT} - Z)) + \mathcal{L}_{\text{MSE}}(\mathbf{F}^T, \mathbf{F}(Z^{T\_GT} - Z)), \tag{2}$$

$$\mathcal{L}_{\text{echo}}(\phi) = \frac{1}{T}\sum_{k=1}^{T}\Big[\mathcal{L}_{\text{CE}}(\mathbf{I}^k, \mathbf{I}(Z^{k\_GT} - Z^{k-1})) + \mathcal{L}_{\text{MSE}}(\mathbf{F}^k, \mathbf{F}(Z^{k\_GT} - Z^{k-1}))\Big], \tag{3}$$

where $\mathbf{I}^k = \mathbf{I}(Z^k - Z^{k-1}), \mathbf{F}^k = \mathbf{F}(Z^k - Z^{k-1})$ with $\mathbf{I}(\cdot)$ and $\mathbf{F}(\cdot)$ representing the extraction of integer-type and float-type contents from the target, respectively. $Z$ denotes the observation obtained by $agent_i$ at time $t$ (omitted in the formula). $Z^{k\_GT}$ denotes the ground truth after $k$ compensation steps of the delayed observation. $\phi$ denotes the model parameters of the compensator.

### 4.3 Delay-reconciled Critic and Curriculum Learning Actor

Wang et al. [34] first introduced the delay-reconciled concept to address single-agent delayed observation problems. Their key insight was that feeding the critic with delay-free global states during centralized training could mitigate the impact of delays. Since the critic's involvement is unnecessary during model inference, the delay-reconciled critic can seamlessly integrate with the centralized training with decentralized execution (CTDE) paradigm.

In addition to receiving compensated observation, the actor may exhibit poor convergence when facing complex scenarios. This occurs because the online training of the compensator also requires time, and the compensator's outputs during the early training stages may significantly deviate from those observed under delay-free conditions. Empirically, the exploration phase in early reinforcement learning training typically plays a crucial role, even though the reward values may not show noticeable improvement during this period. However, unlike the critic, which only operates during training, the final model must ultimately rely on realistically available observation as input. Curriculum learning [3] provides a solution to this challenge. During the initial training phase, we provide the actor with

delay-free observation (i.e., the compensator's ground truth) and gradually reduce the probability of using delay-free observation as training progresses, until the actor relies entirely on compensated observation. A linear annealing strategy is used in our experiments, though more sophisticated or adaptive annealing approaches are also permissible. For less complex tasks (e.g., MPE), actor curriculum learning is not strictly necessary.

### 4.4 Knowledge Distillation

Despite all design and optimization efforts, a discrepancy between compensated observation and delay-free ground truth inevitably persists. However, agents' policies in identical scenarios can be objectively evaluated through quantitative metrics. When training with delayed observation, guidance from a high-performance policy model can steer the model toward more accurate policy optimization or value estimation, thereby accelerating convergence. Motivated by this insight, we incorporate knowledge distillation [16] into the RDC framework. While numerous distillation approaches exist, we experimentally identify an effective methodology. Before training the target model in high-delay environments, we first train a teacher model under low-delay conditions. As anticipated, the teacher model achieves performance closer to ideal delay-free

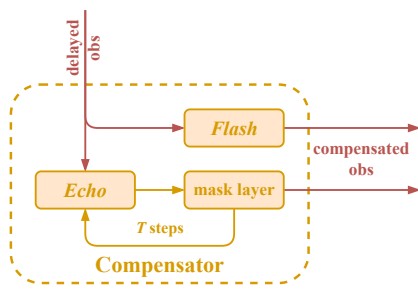

Figure 5: Workflow of *Flash* and *Echo*.

training than independently trained student models in high-delay scenarios. During student model training, we feed the teacher model with compensated observation and employ it to guide both the hidden representations and output decisions of the student actor and critic. The corresponding loss function is formulated as:

$$\mathcal{L}_{kd}(\theta_s) = \mathcal{L}_{\mathrm{CE}}(action_t, action_s) + \beta_1 \cdot \mathcal{L}_{\mathrm{MSE}}(Q_t, Q_s) + \beta_2 \mathcal{L}_{\mathrm{MSE}}(\theta_t^c, \theta_s^c), \quad (4)$$

$$\mathcal{L}_{rdc\_rl}(\theta_s) = \alpha \mathcal{L}_{rl}(\theta_s) + (1 - \alpha)\mathcal{L}_{kd}(\theta_s), \quad (5)$$

where $\beta_1$ and $\beta_2$ denote the weighting factors within the knowledge distillation loss function, and $\alpha$ determines the relative weighting between the knowledge distillation loss and the reinforcement learning loss.

We do not apply knowledge distillation to the compensator or load the teacher's compensator during student training. This design choice stems from two key considerations: First, in online learning scenarios, the compensator's performance should evolve alongside policy improvements, and directly transferring the teacher's compensator knowledge through distillation may not necessarily benefit student policy training. Second, this approach allows for a more explicit demonstration of the effects of pure policy guidance, which we elaborate on in the experimental section. Following a similar principle to curriculum learning, we implement an identical annealing strategy for the knowledge distillation process.

## 5 Experiments

**Scenario Selection:** We select two of the most popular multi-agent reinforcement learning environments for our experiments—MPE[22] and SMAC [28]. Overall, SMAC tasks present greater challenges than MPE tasks due to their more complex state and action spaces. We chose simple-tag (TAG), simple-spread (SPREAD), and simple-reference (REFERENCE) in MPE and three progressively harder scenarios on SMAC: 3s_vs_5z,

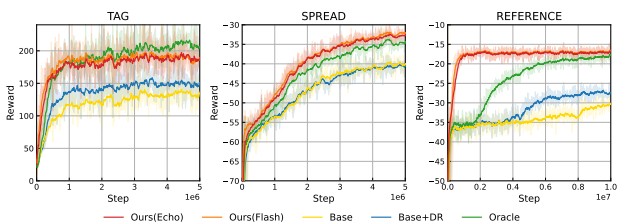

Figure 6: Training performance on MPE.

5m_vs_6m, and 6h_vs_8z. Both benchmarks are discrete environments, so we incorporated value function factorization algorithms into the RDC framework. The reward value obtained per task in

MPE is the sole evaluation metric. While SMAC also provides specific reward values, the win rate is a more critical metric since the objective is to achieve victory.

**Experimental Scheme:** We reformulate the delayed observation problem in MASs using DSID-POMDP and extend it to commonly used benchmarks. Consequently, we implement previous solutions within the RDC framework, such as the delay-reconciled critic and augmented observation input, to enable more equitable comparisons. For baseline RL algorithms, we select code-level optimized FT-QMIX and FT-VDN [17] that demonstrate

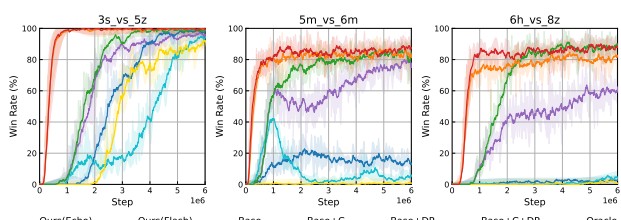

Figure 7: Training performance on SMAC.

significant improvements over vanilla QMIX and VDN, whose performance on discrete tasks has been repeatedly validated [18, 10]. To simplify the presentation, the following abbreviations will be used in the experimental results section: Oracle, Base, Echo, Flash, DR (delay-reconciled critic), H (history input), C (curriculum learning actor), and KD (knowledge distillation). Only the Oracle is trained using the baseline algorithm in a delay-free environment, obtaining the ideal performance that represents the algorithm's capability, unaffected by delays. In the main text, all presented results are based on using FT-QMIX as the baseline algorithm and implementing the compensator with Transformer networks.

**Test Setting:** For every 10,000 training steps, we conduct a test with either 64 or 32 episodes and record model performance. After model training, we perform extensive testing under fixed and unfixed delay conditions, running 1,280 episodes for each setting.

### 5.1 Training

The training performances of FT-QMIX under delayed observation (Base) in Figure 6 and Figure 7 reveal severe performance degradation across all six scenarios, particularly in the 5m_vs_6m and 6h_vs_8z tasks, where win rates approach zero. Our ablation studies employing curriculum learning (Base+C) and delay-reconciled training (Base+DR) without compensation mechanisms show that in complex scenarios, neither curriculum learning alone nor simply providing delay-free states to the critic can satisfactorily counteract the detrimental effects of delayed

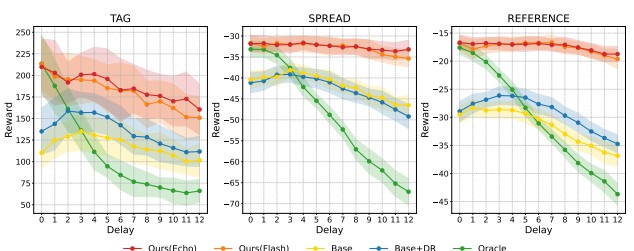

Figure 8: Performance under fixed delay on MPE.

observation on the learning process. These findings suggest that delayed observation has a fundamental impact on both initial convergence and overall policy optimization.

The RDC-enhanced models exhibit significantly faster convergence in most scenarios, demonstrating that while the knowledge distillation process requires sequential training of teacher and student models, it incurs minimal additional training overhead. This efficiency advantage arises from accelerated training in low-delay conditions, where the student model requires fewer than one-third of the original training iterations. Notably, the enhanced models achieve perfor-

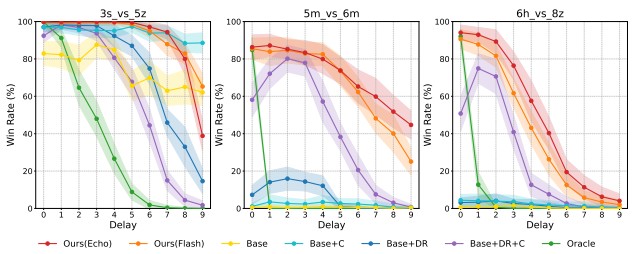

Figure 9: Performance under fixed delay on SMAC.

mance that matches or slightly exceeds that of the delay-free Oracle. The marginal improvement is attributable to additional training steps, as evidenced by Oracle's ongoing performance gains at the end of training in both SPREAD and REFERENCE scenarios.

## 5.2 Performances on different delay settings

Fixed delay testing enables more precise observation of progressive performance degradation and a more accurate assessment of model generalization. As shown in Figure 8 and Figure 9, the Oracle method exhibits substantial performance deterioration as delays increase. The performance drop in the Base and Base+DR methods at delays of 0-2 reveals their inability to generalize to unseen delay conditions during testing, despite successful convergence during training. In contrast, RDC-enhanced models maintain superior performance across all scenarios, demonstrating particularly robust delay adaptation in SPREAD and REFERENCE tasks where reward stability persists despite increasing delays. The model's generalizability on SMAC is significantly lower than on MPE, which indicates that the impact of delay is more pronounced in complex scenarios.

We evaluate model performance under two unfixed delay conditions: in-distribution (within trained delay ranges) and half-out-of-distribution (novel delay ranges). Unless otherwise specified, the random delays in this paper follow a uniform distribution within a given range. The performance under different delay distributions will be discussed later. As shown in Figure 10, RDC-enhanced models with Transformer-based compensators demonstrate only marginal

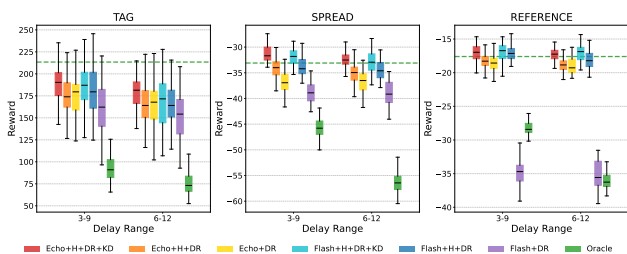

Figure 10: Performance under unfixed delay on MPE.

performance degradation in out-of-distribution tests while maintaining near-Oracle performance (green dashed line). Ablation studies reveal the contribution of each module: Flash+DR underperforms due to the limited input information, while Flash+H+DR shows significant improvement by incorporating history observations. *Echo*'s autoregressive design provides richer input information, resulting in smaller gains from historical data. Knowledge distillation using a low-delay teacher model proves effective, though we excluded Flash+DR from this approach due to its poor baseline performance.

To demonstrate that the RDC framework can adapt to different delay distributions, we evaluate the performance of models trained under a uniform delay distribution when tested on other distributions. As shown in Figure 11, we select the binomial distribution, the normal distribution, and the Poisson distribution as additional evaluation options. In these tests, the delay values are constrained to either the range of 3–9 or 6–12. The binomial distribution, which characterizes

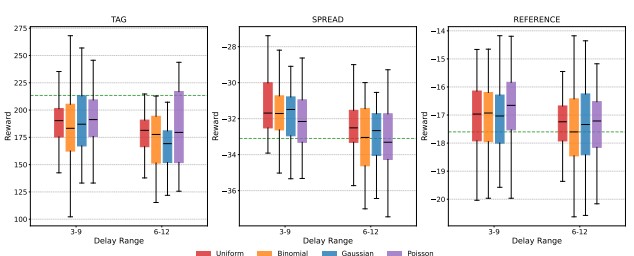

Figure 11: Performance under different delay distributions.

the number of successes in n independent trials, naturally fits our delay settings after a simple shift. Although the values generated by the Poisson distribution are also discrete, they require truncation due to the distribution's unbounded support. For the normal distribution, truncation is typically followed by rounding due to its continuous nature. Considering the properties of different distributions, we adopt the following parameter settings in our experiments: Binomial(6, 0.5) and Binomial(9, 0.5); Poisson(6) and Poisson(9); Normal(6, 2) and Normal(9, 2). The results demonstrate that the model generalizes effectively across various delay distributions. This result is consistent with our expectation, as the compensation process for delayed observations in the RDC framework does not rely on any prior knowledge of the delay distribution. Since different distributions entail different frequencies of high and low delay values, minor performance fluctuations are understandable.

## 5.3 Additional results and analysis

How does the performance of *Flash* compare with that of *Echo*? The results show that as the delay magnitude increases, *Flash* typically demonstrates more pronounced performance degradation than *Echo*, particularly when handling out-of-distribution delays. As illustrated in Figure 12, maintaining identical input history sequence lengths and training configurations is essential for achieving optimal performance with the *Flash* compensator. This suggests an overfitting

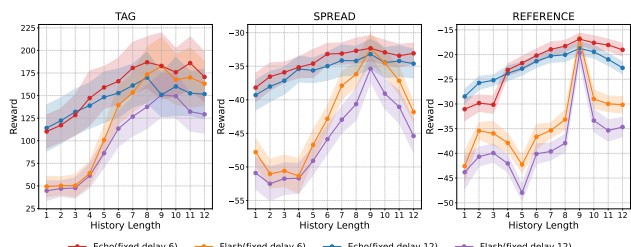

Figure 12: Performance under different history lengths.

compensation pattern, confirming our earlier concern about *Flash*. Although this limitation in generalizability and flexibility is undesirable, we must emphasize its significant advantages in training efficiency and inference speed, which can be decisive factors in specific scenarios. In TAG scenario with a fixed delay of 6, the compensator inference times for *Echo* and *Flash* are approximately 0.02 seconds and 0.004 seconds, respectively.

Why does our method outperform the delay-free Oracle baseline across multiple tests? We align the number of environment steps required for training across all algorithms. As a result, the RDC-enhanced method, which leverages knowledge distillation, obtains additional guidance from the teacher model within the same training horizon. To validate this explanation, we conduct an experiment on MPE where the RDC-enhanced models remained unchanged, while Oracle, Base, and

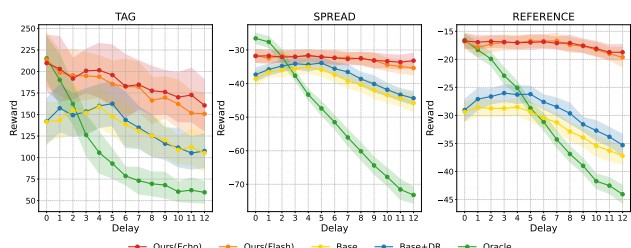

Figure 13: Performance under longer training steps.

Base-DR are trained for an additional 10 million steps, which is the same number of steps used to pretrain the low-delay teacher model. The results in Figure 13 confirm our hypothesis: in the zero-delay setting, the Oracle performance surpasses that of the RDC-enhanced method, consistent with its role as the ideal upper bound. In delayed settings, however, the non-RDC-enhanced methods still suffer from significant performance degradation.

All experimental results are provided in Appendix D, including comprehensive ablation studies and evaluations of different baseline algorithms and compensator network architectures. These experimental results not only validate the compatibility of the RDC framework with algorithms beyond the actor–critic paradigm but also highlight that the choice of baseline algorithm and compensator architecture plays a crucial role in achieving superior performance.

## 6 Conclusion

In this paper, we propose the RDC framework and demonstrate the effectiveness of its constituent modules in addressing delayed observation in MASs. The compensator, as the core component of the framework, directly attempts to reconstruct delay-free observation. The curriculum learning actor and delay-aware critic provide higher-quality data, while knowledge distillation from a low-delay teacher offers policy guidance during model training. The integrated algorithm, which incorporates these modules, achieves outstanding performance with fixed and unfixed delays comparable to those in delay-free environments. Our future research will focus on designing more effective compensator architectures and knowledge distillation techniques to enhance the model's generalizability under varying delays in complex scenarios, as well as developing theoretical frameworks with weaker assumptions. Overall, our research not only presents a multi-agent reinforcement learning algorithm capable of combating delayed observation but also provides a practical training framework, establishing a solid foundation for future studies.

## Acknowledgements

This research is supported by the Oriented Project Independently Deployed by the Institute of Acoustics, Chinese Academy of Sciences (MBDX202402).

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

## Technical Appendices and Supplementary Material

## A Proofs

Here we prove the correctness of the state transition function and observation function expressions defined in DSID-POMDP. First, the state x contains T history states:

$$\mathbf{x} = \{s, s_{(-1)}, ..., s_{(-T)}\}.$$

Expand the expression of the new state transition function:

$$\mathcal{P}_{\mathcal{D}}\left(\mathbf{x}'|\mathbf{x}, \mathbf{a}\right) = \mathcal{P}_{\mathcal{D}}\left(s', s'_{(-1)}, ..., s'_{(-T)}|s, s_{(-1)}, ..., s_{(-T)}, \mathbf{a}\right).$$

Observing these two sequences reveals the transition process of history states $s'_{(-t)} = s_{(-t+1)}$, which can consequently be decomposed into current state transitions and history state transitions. Due to the Markov property, the oldest state in the original state can be directly discarded.

$$\mathcal{P}_{\mathcal{D}}\left(\mathbf{x}'|\mathbf{x}, \mathbf{a}\right) = \mathcal{P}_{\mathcal{D}}\left(s', s'_{(-1)}, ..., s'_{(-T)}|s, s_{(-1)}, ..., s_{(-T)}, \mathbf{a}\right)$$
$$= \mathcal{P}(s'|s, \mathbf{a}) \prod_{t=1}^{T} \delta\left(s'_{(-t)} - s_{(-t+1)}\right)$$

In other words, when the transition of the historical sequence is determined, the state transition function of DSID-POMDP becomes identical to that of POMDP. A fundamental assumption for the validity of the observation function is that observations from different entities are mutually independent. While this assumption holds in most scenarios, it may not be satisfied in specific cases, such as in single-channel communication systems where signals from different entities can interfere. For $agent_i$, if the observation delay value corresponding to $entity_j$ is $t$, it can be expressed as $z^{ij} = s^{ij}_{(-t)}$. Therefore, the probability of this agent obtaining observation $z^{ij}$ from entity $entity_j$ is:

$$\mathcal{O}_{\mathcal{D}}\left(z^{ij}|\mathbf{x}\right) = \sum_{t=0}^{T} p\left(d^{ij} = t\right) \delta\left(z^{ij} - s^{ij}_{(-t)}\right) \mathcal{O}\left(o^{ij} = z^{ij}|s_{(-t)}\right),$$

where $\mathcal{O}\left(o^{ij} = z^{ij}|s_{(-t)}\right)$ denotes the probability of obtaining observation $z^{ij}$ given state $s_{(-t)}$ in the original POMDP. Due to the independence assumption, this definition can be extended to joint observation through factorization:

$$\mathcal{O}_{\mathcal{D}}\left(\mathbf{z}|\mathbf{x}\right) = \prod_{i \in \mathcal{I}} \prod_{j \in \mathcal{I}_{\mathcal{D}}} \sum_{t=0}^{T} p\left(d^{ij} = t\right) \delta\left(z^{ij} - s^{ij}_{(-t)}\right) \mathcal{O}\left(o^{ij} = z^{ij}|s_{(-t)}\right).$$

## B Scenario Introduction

**simple-tag**: This is a competitive predator-prey simulation where good agents must evade slower but aggressive adversaries. The good agents are faster but incur a penalty for each collision with an adversary, while adversaries are rewarded for successfully hitting them. The terrain includes static obstacles that block movement. To prevent good agents from escaping indefinitely, they are penalized for exiting the designated area based on a predefined boundary function. By default, the scenario starts with one good agent, three adversaries, and two obstacles, creating a dynamic balance of pursuit and evasion. In our experiments, the policy of good agents is fixed as a pre-trained model with MADDPG [22], consistent with Papoudakis et al. [26]. The algorithm only controls the adversaries, transforming the adversarial environment into a cooperative one.

**simple-spread**: In this cooperative multi-agent scenario, N agents (default: 3) must learn to cover N landmarks efficiently while avoiding collisions. The agents are collectively rewarded based on how well they cover all landmarks, measured by the sum of the minimum distances between each landmark and its closest agent. Agents must minimize this global distance metric to maximize their shared reward. To prevent reckless behavior, each agent is individually penalized for colliding with other agents. The trade-off between global coverage and collision avoidance is adjustable via the $local\_ratio$ parameter, enabling fine-tuned control over agent coordination strategies.

Table 1: Mean Environment parameters.

| MPE | | SMAC | |
|---|---|---|---|
| Parameter | Value | Parameter | Value |
| time_limit | 25 | difficulty | 7 |
| obs_agent_id | True | obs_agent_id | True |
| obs_last_action | False | obs_last_action | True |
| | | state_last_action | True |
| | | state_timestep_number | False |
| | | conic_fov | False |

**simple-reference**: This scenario involves two cooperative agents navigating toward three uniquely colored landmarks, each with a hidden target assignment. Crucially, an agent's target landmark is only known by the other agent, requiring real-time communication to resolve uncertainty. Both agents act as speakers and listeners, exchanging information to infer each other's goals while navigating.

**SMAC**: Unlike MPE, the objective in all SMAC scenarios is to achieve victory, with the sole approach being the complete elimination of enemy forces. The scenario names explicitly indicate the force compositions of both allied and enemy units—for example, 6h_vs_8z denotes six Hydralisks battling eight Zealots. This environment emphasizes short-term coordination among agents and the effective utilization of unit-specific advantages.

# C   Implementation Details

## C.1   Delayed Observation Implementation

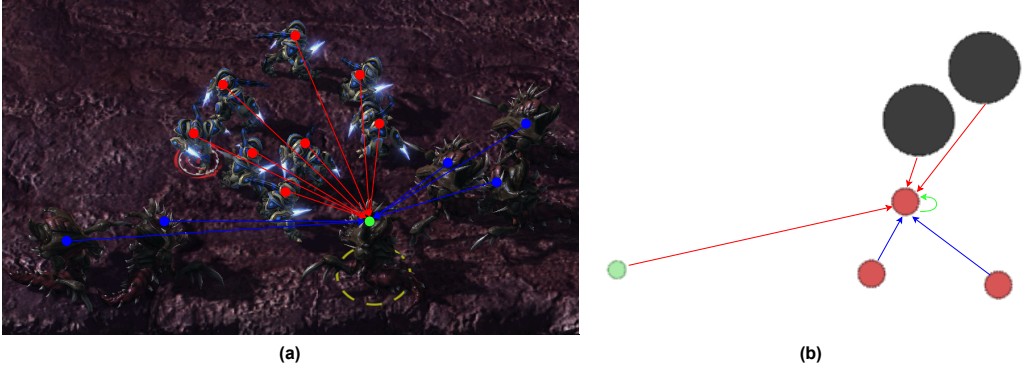

Figure 14: Applications of DSID-POMDP on SMAC's 6h_vs_8z and MPE's simple-tag.

We develop delay filters for SMAC and MPE to simulate realistic perception and communication delays, implementing four distinct modes: no delay (none), fixed delay (f), partially fixed delay (pf), and unfixed delay (uf). The fixed delay mode sets a fixed delay value and applies it to all observations. The partially unfixed delay mode uses fixed delay values but does not necessarily delay all observation contents. The unfixed delay mode introduces randomness in both delay values and which contents get delayed. While delay filters support different distributions or distance-based delay calculations between entities and agents, our experiments use uniform distributions for all random variables, without loss of generality. The delay implementation mechanism primarily maintains history observation records for each agent and retrieves information from past observations at the current timestep according to specific delay policies.

We classify agent observation into four categories based on their delay characteristics: movement features (real-time updated information), enemy features (environmental entity data excluding allies), ally features (information from allied agents), and self features (agent's state). In practice, naming these features is not crucial—our primary distinction lies in their delay characteristics. For example, in SMAC's 6h_vs_8z scenario, observations include movement capabilities (4 dimensions), 8 enemy states (each with 6 dimensions), 5 ally statuses (each with 5 dimensions), and self-attributes (1 dimension). Our implementation delays only enemy and ally features, while maintaining real-time

updates for movement and self features. This reflects real-world conditions, where agents have immediate access to their own states but experience delayed environmental perception.

The system initializes the observation history at the start of each episode and continuously records delay-free observations. It retrieves historical data based on configured delay parameters when generating delayed observations. For example, in MPE's simple-tag scenario, predators may track evaders using positions delayed by 2 timesteps, impairing pursuit efficiency. Similarly, in SMAC, troops exhibit response lags to enemy movements. The system maintains temporal consistency by ensuring delayed observations are always drawn from valid history states while preserving the original environment dynamics. This implementation enables systematic investigation of various delay patterns (fixed, unfixed, or distance-dependent) while maintaining the integrity of the underlying multi-agent decision-making process. The modular design allows for the flexible configuration of delay parameters without modifying the core environment logic.

Additionally, the delay filter enforces temporal consistency constraints, ensuring that information observed at step $t$ cannot be older than that at step $t - 1$. When the delay-reconciled critic is not employed, the global state is formed by concatenating observations from all agents. By implementing delay filters in existing simulation environments, we establish a reliable and flexible experimental platform for investigating the impact of delayed observation on the performance of MASs.

## C.2   Compensator Implementation

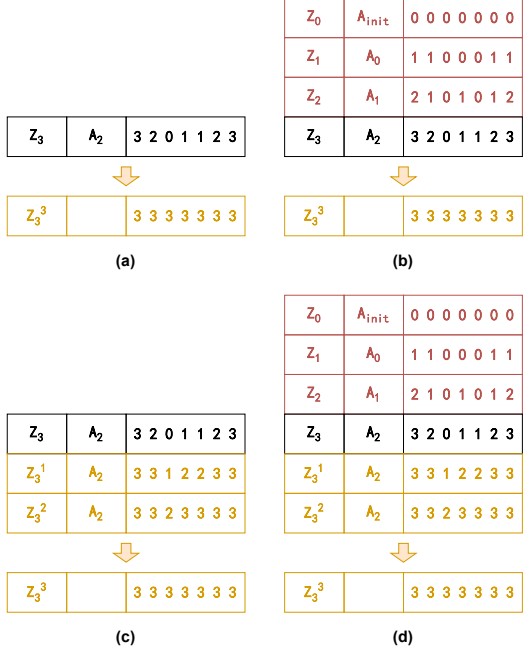

Figure 15: Inputs and outputs of compensators. (a): *Flash* without history inputs. (b): *Flash* with history inputs. (c): *Echo* without history inputs. (d): *Echo* with history inputs.

We utilize a deep learning-based compensator to mitigate the effects of observation delay in MASs, and employ GRU-based and Transformer-based architectures to predict delay-free observation from historical data. The compensator implementation consists of input sequence construction, label generation, and mask generation. The compensator processes input sequences constructed from past observations and actions, with optional T-step historical context (padded when insufficient). Figure 15 follows the example from Figure 3. Both Flash+H and Echo+H models acquire additional information. Notably, *Echo*'s autoregressive inference aligns closely with historical sequence processing, which helps the model understand the relationships between delay and observation. The figure visually represents delay value sequences as their corresponding actual environment timestep sequences. In practical implementation, we input delay value sequences for *Flash* and binary (0 or 1) sequences indicating the presence of delay for *Echo*.

Table 2: Mean Training parameters.

| MPE | | SMAC | |
|---|---|---|---|
| Parameter | Value | Parameter | Value |
| t_max | 5e6/1e7 | t_max | 6e6 |
| test_nepisode | 64 | test_nepisode | 32 |
| batch_size | 32 | batch_size | 128 |
| epsilon_anneal_time | 5e4 | epsilon_anneal_time | 1e5/5e6 |
| standardise_rewards | True | standardise_rewards | False |
| actor_model | GRU | actor_model | GRU |
| target_update_interval | 200 | target_update_interval | 200 |
| mixing_embed_dim | 32 | mixing_embed_dim | 32 |
| hypernet_embed | 64 | hypernet_embed | 64 |
| actor_hidden_dim | 64 | actor_hidden_dim | 64 |
| td_lambda | 0.6 | td_lambda | 0.6/0.3 |
| rl_learning_rate | 1e-3 | rl_learning_rate | 1e-3 |
| compensator_model | GRU/Transformer | compensator_model | GRU/Transformer |
| compensator_hidden_dim | 64 | compensator_hidden_dim | 64 |
| compensator_mode | None/Flash/Echo | compensator_mode | None/Flash/Echo |
| compensator_learning_rate | 1e-3 | compensator_learning_rate | 1e-3 |
| delay_type | unfixed | delay_type | unfixed |
| delay_value | 6 | delay_value | 3 |
| delay_scope | 3 | delay_scope | 3 |
| use_history | True/False | use_history | True/False |
| history_length | 9 | history_length | 6 |
| delay_reconciled | True/False | delay_reconciled | True/False |
| curriculum_start_value | 0 | curriculum_start_value | 1/0 |
| curriculum_end_value | 0 | curriculum_end_value | 0 |
| curriculum_start_step | 1e6 | curriculum_start_value | 1e6 |
| curriculum_end_step | 3e6 | curriculum_end_step | 4e6 |
| distillation_start_value | 1/0 | distillation_start_value | 1/0 |
| distillation_end_value | 0 | distillation_end_value | 0 |
| distillation_start_step | 2e6/3e6 | distillation_start_value | 2e6 |
| distillation_end_step | 4e6/7e6 | distillation_end_step | 4e6 |

We adopt a supervised learning framework where the label generation module creates ground truth from stored delay-free observations, using ideal delay values as reference. The mask generation module prevents error propagation in *Echo* by masking previously compensated content. Our hybrid loss function combines mean squared error (MSE) for continuous features (e.g., positions) with weighted cross-entropy (CE) for discrete features (e.g., unit status), balancing their numerical scales. To enhance training efficiency and stability, we implement teacher forcing - initially using real observation as next-step input with 100% probability, gradually decreasing to 0% during training to help the model learn to rely on its predictions. As shown in Table 3, the teacher forcing training mode demonstrates no significant advantage compared to not using teacher forcing. Therefore, we disable this option by default during training.

## C.3   Training Details

In this section, we present the hyperparameter settings for different tasks. Table 1 and Table 2 detail the key environmental and training parameters, respectively. While ensuring algorithm performance, we maintained consistency in hyperparameters as much as possible. All actor networks use the GRU architecture, while QMIX's critic employs a two-layer hypernetwork. The GRU compensator consists of one GRU layer and three linear layers, whereas the Transformer compensator uses only one pair of encoder-decoder layers and supports a pure encoder structure. The replay buffer size is fixed at 5000, with batch sizes set to 32 for MPE and 128 for SMAC. During training initialization, $\epsilon = 1$ and linearly anneals to $\epsilon = 0.05$. The default training step for MPE is set to 5e6. However, we observed a second significant performance surge around 2e6 steps on REFERENCE, with performance still maintaining an upward trend at 5e6 steps. Consequently, we extended the training steps to 1e7 to provide sufficient convergence time for each algorithm. Accordingly, the starting and ending steps for knowledge distillation were also increased.

We do not entirely disregard the curriculum learning option in MPE scenarios. However, early experiments revealed that this technique proved ineffective for MPE, while being indispensable for SMAC. This result demonstrates that curriculum learning actors can significantly mitigate convergence issues caused by random seeds when handling complex tasks. In reinforcement learning, the problem of model non-convergence due to the improper timing of performance ascent is prevalent. Careful

observation of convergence curves reveals that performance ascent typically does not occur during the initial training phase in SMAC tasks. In contrast, in MPE tasks, performance improvement closely follows the start of training. This timing difference may explain why curriculum learning actors succeed remarkably in SMAC scenarios.

The teacher model is trained for 1e7 steps across all tasks with a delay setting of 1-3. We attempted to directly employ the Oracle as the teacher model, where, during student model training, the Oracle receives delay-free observation and provides immediate guidance. This approach proved unsuccessful, likely due to the inherent discrepancy between delay-free observation and compensated observation. Specifically, when the student model receives compensated observation, decisions based on delay-free observation might be objectively superior but could disrupt the student model's judgment. To address this issue, we experiment with a low-delay teacher model, achieving exceptional performance. We encourage other researchers to explore further variations within our framework.

# D  Supplementary Results

Table 3: Performance comparison with teacher forcing enabled and disabled.

| | TF | DR | H | 0 | 3 | 6 | 9 | 12 |
|---|---|---|---|---|---|---|---|---|
| Fixed Delay Value | | | | 0 | 3 | 6 | 9 | 12 |
| | TF | DR | H | | | TAG | | |
| Echo | ✓ | ✓ | ✓ | $190.9 \pm 26.0$ | $175.8 \pm 29.7$ | $168.8 \pm 23.4$ | $161.7 \pm 25.4$ | $158.5 \pm 25.9$ |
| Echo | | ✓ | ✓ | $185.9 \pm 23.4$ | $175.1 \pm 30.3$ | $176.9 \pm 29.9$ | $167.3 \pm 24.0$ | $155.5 \pm 25.4$ |
| | TF | DR | H | | | SPREAD | | |
| Echo | ✓ | ✓ | ✓ | $-34.2 \pm 2.2$ | $-34.1 \pm 1.9$ | $-34.2 \pm 2.0$ | $-35.3 \pm 2.2$ | $-35.4 \pm 2.1$ |
| Echo | | ✓ | ✓ | $-33.6 \pm 1.7$ | $-33.8 \pm 2.2$ | $-34.3 \pm 1.5$ | $-35.4 \pm 1.9$ | $-36.6 \pm 1.7$ |

Table 4: Fixed delay test results (rewards) of MPE with baseline algorithm FT-QMIX.

| | DR | H | KD | 0 | 3 | 6 | 9 | 12 |
|---|---|---|---|---|---|---|---|---|
| Fixed Delay Value | | | | 0 | 3 | 6 | 9 | 12 |
| | DR | H | KD | | | TAG | | |
| Oracle | | | | $\mathbf{213.4 \pm 33.5}$ | $136.2 \pm 23.0$ | $84.5 \pm 16.9$ | $70.1 \pm 11.1$ | $66.1 \pm 13.3$ |
| Base | | | | $110.2 \pm 17.7$ | $135.2 \pm 22.5$ | $125.4 \pm 22.8$ | $112.6 \pm 20.7$ | $101.5 \pm 19.6$ |
| Base | ✓ | | | $135.1 \pm 26.7$ | $156.9 \pm 23.2$ | $142.3 \pm 22.3$ | $120.9 \pm 20.8$ | $111.7 \pm 17.5$ |
| Flash | ✓ | | | $176.6 \pm 27.7$ | $177.5 \pm 34.2$ | $150.4 \pm 22.5$ | $132.8 \pm 21.6$ | $127.8 \pm 23.7$ |
| Flash | ✓ | ✓ | | $188.0 \pm 26.6$ | $180.4 \pm 24.7$ | $166.7 \pm 29.0$ | $168.5 \pm 28.2$ | $149.9 \pm 26.3$ |
| **Flash** | ✓ | ✓ | ✓ | $\mathbf{213.7 \pm 30.8}$ | $\mathbf{194.8 \pm 27.5}$ | $\mathbf{182.5 \pm 32.0}$ | $\mathbf{169.8 \pm 23.9}$ | $\mathbf{150.9 \pm 24.1}$ |
| Echo | ✓ | | | $194.2 \pm 29.0$ | $187.6 \pm 29.0$ | $173.8 \pm 22.6$ | $160.7 \pm 30.1$ | $163.7 \pm 27.5$ |
| Echo | ✓ | ✓ | | $185.9 \pm 23.4$ | $175.1 \pm 30.3$ | $176.9 \pm 29.9$ | $167.3 \pm 24.0$ | $155.5 \pm 25.4$ |
| Echo | ✓ | | ✓ | $206.5 \pm 34.0$ | $197.9 \pm 28.5$ | $184.7 \pm 32.4$ | $175.6 \pm 22.9$ | $165.2 \pm 28.0$ |
| **Echo** | ✓ | ✓ | ✓ | $\mathbf{209.8 \pm 32.3}$ | $\mathbf{200.8 \pm 25.1}$ | $\mathbf{182.9 \pm 36.6}$ | $\mathbf{176.2 \pm 24.8}$ | $\mathbf{160.5 \pm 29.5}$ |
| | DR | H | KD | | | SPREAD | | |
| Oracle | | | | $\mathbf{-33.1 \pm 2.0}$ | $-37.6 \pm 2.2$ | $-48.8 \pm 2.0$ | $-59.9 \pm 2.9$ | $-67.2 \pm 3.2$ |
| Base | | | | $-40.4 \pm 2.2$ | $-38.1 \pm 1.8$ | $-40.3 \pm 2.4$ | $-44.2 \pm 2.2$ | $-46.5 \pm 2.3$ |
| Base | ✓ | | | $-41.1 \pm 2.4$ | $-39.2 \pm 1.8$ | $-41.0 \pm 2.5$ | $-44.7 \pm 2.0$ | $-49.2 \pm 2.9$ |
| Flash | ✓ | | | $-37.7 \pm 1.7$ | $-37.8 \pm 2.3$ | $-39.3 \pm 1.9$ | $-40.1 \pm 2.1$ | $-40.0 \pm 2.3$ |
| Flash | ✓ | ✓ | | $-33.9 \pm 1.8$ | $-34.2 \pm 2.0$ | $-33.1 \pm 1.9$ | $-34.4 \pm 1.8$ | $-35.8 \pm 1.8$ |
| **Flash** | ✓ | ✓ | ✓ | $\mathbf{-31.7 \pm 2.0}$ | $\mathbf{-32.0 \pm 1.6}$ | $\mathbf{-32.4 \pm 2.3}$ | $\mathbf{-33.2 \pm 2.3}$ | $\mathbf{-35.4 \pm 2.0}$ |
| Echo | ✓ | | | $-36.3 \pm 2.2$ | $-36.6 \pm 2.3$ | $-36.2 \pm 1.9$ | $-36.8 \pm 1.7$ | $-37.4 \pm 2.2$ |
| Echo | ✓ | ✓ | | $-33.6 \pm 1.7$ | $-33.8 \pm 2.2$ | $-34.3 \pm 1.5$ | $-35.4 \pm 1.9$ | $-36.6 \pm 1.7$ |
| Echo | ✓ | | ✓ | $-32.2 \pm 1.6$ | $-33.0 \pm 2.3$ | $-33.3 \pm 1.8$ | $-33.3 \pm 2.2$ | $-33.9 \pm 1.9$ |
| **Echo** | ✓ | ✓ | ✓ | $\mathbf{-31.8 \pm 1.9}$ | $\mathbf{-32.0 \pm 2.0}$ | $\mathbf{-32.3 \pm 1.9}$ | $\mathbf{-33.1 \pm 2.1}$ | $\mathbf{-33.2 \pm 2.2}$ |
| | DR | H | KD | | | REFERENCE | | |
| Oracle | | | | $\mathbf{-17.6 \pm 1.2}$ | $-22.5 \pm 1.2$ | $-31.1 \pm 1.1$ | $-38.2 \pm 1.5$ | $-43.7 \pm 1.9$ |
| Base | | | | $-29.5 \pm 1.5$ | $-28.6 \pm 1.5$ | $-30.3 \pm 1.7$ | $-34.3 \pm 1.5$ | $-36.8 \pm 1.8$ |
| Base | ✓ | | | $-28.9 \pm 1.8$ | $-26.1 \pm 1.6$ | $-27.7 \pm 1.7$ | $-30.9 \pm 1.7$ | $-34.7 \pm 1.6$ |
| Flash | ✓ | | | $-34.5 \pm 2.1$ | $-34.2 \pm 2.0$ | $-34.8 \pm 2.1$ | $-34.9 \pm 2.3$ | $-35.1 \pm 2.2$ |
| Flash | ✓ | ✓ | | $-17.1 \pm 1.3$ | $-17.4 \pm 1.5$ | $-17.3 \pm 1.2$ | $-18.0 \pm 1.5$ | $-20.0 \pm 1.2$ |
| **Flash** | ✓ | ✓ | ✓ | $\mathbf{-16.8 \pm 1.3}$ | $\mathbf{-17.0 \pm 1.3}$ | $\mathbf{-16.7 \pm 1.1}$ | $\mathbf{-17.6 \pm 1.1}$ | $\mathbf{-19.6 \pm 1.7}$ |
| Echo | ✓ | | | $-19.1 \pm 1.4$ | $-18.0 \pm 1.4$ | $-19.0 \pm 1.4$ | $-19.0 \pm 1.4$ | $-19.1 \pm 1.6$ |
| Echo | ✓ | ✓ | | $-18.4 \pm 1.2$ | $-18.5 \pm 1.5$ | $-18.5 \pm 1.1$ | $-18.7 \pm 1.4$ | $-20.2 \pm 1.4$ |
| Echo | ✓ | | ✓ | $-20.6 \pm 1.2$ | $-20.9 \pm 1.4$ | $-21.3 \pm 1.3$ | $-22.2 \pm 1.1$ | $-22.2 \pm 1.2$ |
| **Echo** | ✓ | ✓ | ✓ | $\mathbf{-16.7 \pm 1.4}$ | $\mathbf{-16.9 \pm 1.1}$ | $\mathbf{-16.9 \pm 1.2}$ | $\mathbf{-17.6 \pm 1.2}$ | $\mathbf{-18.7 \pm 1.4}$ |

Table 5: Unfixed delay test results (rewards) of MPE with baseline algorithm FT-QMIX.

| Delay Range | DR | H | KD | 3-9 | 6-12 | 3-9 | 6-12 | 3-9 | 6-12 |
|---|---|---|---|---|---|---|---|---|---|
| | | | | TAG | | SPREAD | | REFERENCE | |
| Oracle | | | | $92.3 \pm 15.2$ | $74.8 \pm 13.5$ | $-45.9 \pm 1.8$ | $-56.4 \pm 2.1$ | $-28.3 \pm 1.2$ | $-36.1 \pm 1.4$ |
| Base | | | | $127.0 \pm 18.8$ | $113.6 \pm 19.9$ | $-39.6 \pm 2.0$ | $-42.2 \pm 2.5$ | $-29.9 \pm 1.6$ | $-33.2 \pm 2.0$ |
| Base | ✓ | | | $144.6 \pm 21.1$ | $132.4 \pm 19.7$ | $-40.5 \pm 2.2$ | $-44.0 \pm 2.3$ | $-27.4 \pm 1.6$ | $-30.4 \pm 1.9$ |
| Flash | ✓ | | | $161.5 \pm 29.4$ | $151.4 \pm 27.4$ | $-38.8 \pm 2.3$ | $-39.2 \pm 2.6$ | $-34.9 \pm 2.0$ | $-35.2 \pm 2.0$ |
| Flash | ✓ | ✓ | | $180.3 \pm 28.3$ | $165.4 \pm 26.1$ | $-33.7 \pm 2.0$ | $-34.5 \pm 1.9$ | $-17.1 \pm 1.1$ | $-18.2 \pm 1.3$ |
| **Flash** | ✓ | ✓ | ✓ | $\mathbf{188.6 \pm 31.1}$ | $\mathbf{166.9 \pm 30.9}$ | $\mathbf{-32.0 \pm 1.9}$ | $\mathbf{-33.0 \pm 2.3}$ | $\mathbf{-17.0 \pm 1.4}$ | $\mathbf{-17.2 \pm 1.4}$ |
| Echo | ✓ | | | $175.1 \pm 25.0$ | $165.5 \pm 24.5$ | $-36.7 \pm 2.2$ | $-36.9 \pm 2.2$ | $-18.6 \pm 1.3$ | $-18.9 \pm 1.3$ |
| Echo | ✓ | ✓ | | $177.1 \pm 21.9$ | $166.8 \pm 27.4$ | $-34.2 \pm 2.0$ | $-35.1 \pm 1.9$ | $-18.3 \pm 1.2$ | $-18.9 \pm 1.2$ |
| Echo | ✓ | | ✓ | $178.9 \pm 27.0$ | $180.0 \pm 20.7$ | $-32.1 \pm 2.2$ | $-33.2 \pm 1.9$ | $-21.4 \pm 1.0$ | $-21.6 \pm 1.3$ |
| **Echo** | ✓ | ✓ | ✓ | $\mathbf{189.2 \pm 22.3}$ | $\mathbf{178.3 \pm 21.1}$ | $\mathbf{-31.4 \pm 1.9}$ | $\mathbf{-32.5 \pm 1.4}$ | $\mathbf{-17.0 \pm 1.2}$ | $\mathbf{-17.4 \pm 1.1}$ |

Table 6: Fixed delay test results of SMAC (win rates) with baseline algorithm FT-QMIX.

| | C | DR | H | KD | 0 | 2 | 4 | 6 | 8 |
|---|---|---|---|---|---|---|---|---|---|---|
| | | | | | 3s_vs_5z | | | | |
| Oracle | | | | | $\mathbf{99.7 \pm 1.2}$ | $64.6 \pm 9.3$ | $26.6 \pm 7.9$ | $1.9 \pm 2.4$ | $0.1 \pm 0.5$ |
| Base | | | | | $82.9 \pm 6.2$ | $79.3 \pm 8.1$ | $84.9 \pm 6.1$ | $69.9 \pm 9.1$ | $65.0 \pm 9.1$ |
| Base | ✓ | | | | $97.0 \pm 2.6$ | $95.5 \pm 2.9$ | $95.1 \pm 3.6$ | $93.8 \pm 4.5$ | $88.4 \pm 4.8$ |
| Base | | ✓ | | | $97.0 \pm 2.7$ | $98.0 \pm 2.0$ | $92.3 \pm 5.8$ | $74.8 \pm 8.3$ | $33.0 \pm 10.5$ |
| Base | ✓ | ✓ | | | $92.4 \pm 3.6$ | $96.8 \pm 2.8$ | $80.6 \pm 5.9$ | $44.5 \pm 8.9$ | $4.5 \pm 3.4$ |
| Flash | ✓ | ✓ | | | $95.0 \pm 4.3$ | $96.6 \pm 3.0$ | $91.4 \pm 5.0$ | $78.1 \pm 6.2$ | $37.7 \pm 10.2$ |
| Flash | ✓ | ✓ | ✓ | | $98.8 \pm 2.1$ | $97.3 \pm 2.9$ | $93.4 \pm 4.5$ | $83.6 \pm 6.5$ | $43.5 \pm 7.9$ |
| **Flash** | | ✓ | ✓ | ✓ | $\mathbf{99.8 \pm 0.7}$ | $\mathbf{99.3 \pm 1.9}$ | $\mathbf{99.4 \pm 1.2}$ | $\mathbf{94.9 \pm 3.4}$ | $\mathbf{82.9 \pm 5.4}$ |
| Echo | ✓ | ✓ | | | $96.3 \pm 2.8$ | $97.1 \pm 2.7$ | $94.6 \pm 3.8$ | $83.6 \pm 6.3$ | $24.7 \pm 8.5$ |
| Echo | ✓ | ✓ | ✓ | | $98.5 \pm 2.2$ | $98.8 \pm 1.8$ | $96.6 \pm 2.8$ | $84.2 \pm 6.7$ | $60.9 \pm 8.0$ |
| **Echo** | | ✓ | ✓ | ✓ | $\mathbf{99.8 \pm 0.8}$ | $\mathbf{99.8 \pm 0.7}$ | $\mathbf{99.8 \pm 0.8}$ | $\mathbf{97.1 \pm 2.7}$ | $\mathbf{80.0 \pm 8.1}$ |
| | C | DR | H | KD | 5m_vs_6m | | | | |
| Oracle | | | | | $\mathbf{84.8 \pm 6.7}$ | $0.0 \pm 0.0$ | $0.0 \pm 0.0$ | $0.0 \pm 0.0$ | $0.0 \pm 0.0$ |
| Base | | | | | $0.5 \pm 1.1$ | $0.4 \pm 1.0$ | $0.9 \pm 1.4$ | $0.5 \pm 1.3$ | $0.1 \pm 0.5$ |
| Base | ✓ | | | | $1.1 \pm 1.6$ | $2.7 \pm 2.9$ | $3.4 \pm 3.5$ | $2.3 \pm 2.8$ | $0.7 \pm 1.6$ |
| Base | | ✓ | | | $7.3 \pm 5.1$ | $15.9 \pm 6.2$ | $12.1 \pm 5.4$ | $0.4 \pm 1.0$ | $0.1 \pm 0.5$ |
| Base | ✓ | ✓ | | | $58.1 \pm 9.0$ | $80.2 \pm 7.1$ | $57.2 \pm 10.8$ | $20.5 \pm 8.1$ | $3.0 \pm 2.8$ |
| Flash | ✓ | ✓ | | | $83.2 \pm 6.4$ | $80.6 \pm 7.4$ | $73.7 \pm 7.7$ | $59.9 \pm 9.5$ | $36.6 \pm 8.4$ |
| Flash | ✓ | ✓ | ✓ | | $84.8 \pm 4.4$ | $81.9 \pm 6.4$ | $78.4 \pm 6.4$ | $43.9 \pm 7.8$ | $5.8 \pm 4.3$ |
| **Flash** | | ✓ | ✓ | ✓ | $\mathbf{85.5 \pm 5.7}$ | $\mathbf{84.6 \pm 6.8}$ | $\mathbf{82.5 \pm 6.1}$ | $\mathbf{62.3 \pm 7.7}$ | $\mathbf{40.1 \pm 9.8}$ |
| Echo | ✓ | ✓ | | | $81.9 \pm 7.2$ | $78.4 \pm 7.1$ | $74.7 \pm 7.3$ | $58.4 \pm 10.2$ | $44.7 \pm 8.3$ |
| Echo | ✓ | ✓ | ✓ | | $73.4 \pm 7.1$ | $65.2 \pm 7.9$ | $63.1 \pm 9.8$ | $39.8 \pm 7.8$ | $23.4 \pm 7.7$ |
| **Echo** | | ✓ | ✓ | ✓ | $\mathbf{86.3 \pm 6.5}$ | $\mathbf{85.2 \pm 5.0}$ | $\mathbf{79.9 \pm 7.7}$ | $\mathbf{65.2 \pm 8.9}$ | $\mathbf{51.8 \pm 8.6}$ |
| | C | DR | H | KD | 6h_vs_8z | | | | |
| Oracle | | | | | $\mathbf{92.0 \pm 4.0}$ | $0.5 \pm 1.2$ | $0.0 \pm 0.0$ | $0.0 \pm 0.0$ | $0.0 \pm 0.0$ |
| Base | | | | | $0.6 \pm 1.2$ | $1.0 \pm 1.9$ | $0.7 \pm 1.5$ | $0.1 \pm 0.5$ | $0.0 \pm 0.0$ |
| Base | ✓ | | | | $4.4 \pm 3.1$ | $3.8 \pm 3.5$ | $2.4 \pm 2.6$ | $1.4 \pm 1.8$ | $0.7 \pm 1.5$ |
| Base | | ✓ | | | $3.1 \pm 2.5$ | $4.1 \pm 3.8$ | $2.1 \pm 2.7$ | $0.4 \pm 1.0$ | $0.2 \pm 1.0$ |
| Base | ✓ | ✓ | | | $50.8 \pm 9.1$ | $70.6 \pm 9.2$ | $12.7 \pm 6.1$ | $2.7 \pm 3.5$ | $0.3 \pm 0.9$ |
| Flash | ✓ | ✓ | | | $77.5 \pm 6.6$ | $70.7 \pm 9.0$ | $29.7 \pm 10.9$ | $4.6 \pm 3.6$ | $0.5 \pm 1.4$ |
| Flash | ✓ | ✓ | ✓ | | $85.3 \pm 7.1$ | $72.1 \pm 8.6$ | $30.6 \pm 7.9$ | $5.9 \pm 4.4$ | $1.2 \pm 2.4$ |
| **Flash** | | ✓ | ✓ | ✓ | $\mathbf{90.6 \pm 5.1}$ | $\mathbf{81.6 \pm 6.1}$ | $\mathbf{43.1 \pm 8.7}$ | $\mathbf{12.6 \pm 6.1}$ | $\mathbf{3.4 \pm 3.4}$ |
| Echo | ✓ | ✓ | | | $86.6 \pm 7.2$ | $82.0 \pm 7.3$ | $40.8 \pm 9.6$ | $11.2 \pm 5.6$ | $2.9 \pm 3.1$ |
| Echo | ✓ | ✓ | ✓ | | $86.7 \pm 6.2$ | $71.0 \pm 8.2$ | $19.1 \pm 6.8$ | $5.1 \pm 4.3$ | $0.6 \pm 1.2$ |
| **Echo** | | ✓ | ✓ | ✓ | $\mathbf{94.1 \pm 4.1}$ | $\mathbf{89.3 \pm 6.3}$ | $\mathbf{57.6 \pm 10.7}$ | $\mathbf{19.5 \pm 6.7}$ | $\mathbf{6.2 \pm 5.1}$ |

Table 7: Unfixed delay test results (win rates) of SMAC with baseline algorithm FT-QMIX.

| Delay Range | C | DR | H | KD | 0-6 | 3-9 | 0-6 | 3-9 | 0-6 | 3-9 |
|---|---|---|---|---|---|---|---|---|---|---|---|
| | | | | | 3s_vs_5z | | | | | |
| Oracle | | | | | $62.2 \pm 7.8$ | $16.2 \pm 6.4$ | $0.0 \pm 0.0$ | $0.0 \pm 0.0$ | $2.9 \pm 3.2$ | $0.0 \pm 0.0$ |
| Base | | | | | $83.4 \pm 6.5$ | $77.5 \pm 6.4$ | $0.7 \pm 1.3$ | $0.5 \pm 1.4$ | $0.5 \pm 1.2$ | $0.4 \pm 1.0$ |
| Base | ✓ | | | | $95.9 \pm 3.1$ | $94.8 \pm 3.7$ | $2.1 \pm 2.1$ | $2.0 \pm 2.3$ | $5.5 \pm 3.9$ | $1.3 \pm 1.7$ |
| Base | | ✓ | | | $98.1 \pm 1.7$ | $84.8 \pm 6.2$ | $15.2 \pm 7.6$ | $4.1 \pm 3.2$ | $2.3 \pm 2.7$ | $0.9 \pm 1.4$ |
| Base | ✓ | ✓ | | | $97.0 \pm 2.6$ | $74.7 \pm 7.4$ | $79.0 \pm 8.4$ | $45.4 \pm 8.5$ | $68.6 \pm 11.8$ | $7.2 \pm 5.4$ |
| Flash | ✓ | ✓ | | | $96.5 \pm 3.7$ | $92.4 \pm 5.4$ | $78.5 \pm 6.3$ | $71.1 \pm 7.8$ | $67.1 \pm 8.1$ | $13.7 \pm 6.6$ |
| Flash | ✓ | ✓ | ✓ | | $97.2 \pm 2.9$ | $90.6 \pm 4.8$ | $83.7 \pm 6.6$ | $73.5 \pm 7.8$ | $73.8 \pm 6.4$ | $22.3 \pm 8.1$ |
| **Flash** | | ✓ | ✓ | ✓ | $\mathbf{99.7 \pm 0.9}$ | $\mathbf{98.2 \pm 2.6}$ | $\mathbf{83.0 \pm 7.0}$ | $\mathbf{76.6 \pm 7.6}$ | $\mathbf{80.7 \pm 7.1}$ | $\mathbf{28.9 \pm 7.6}$ |
| Echo | ✓ | ✓ | | | $98.9 \pm 2.0$ | $95.5 \pm 3.8$ | $80.2 \pm 7.8$ | $66.9 \pm 8.6$ | $78.8 \pm 7.1$ | $27.5 \pm 9.4$ |
| Echo | ✓ | ✓ | ✓ | | $98.9 \pm 1.6$ | $96.2 \pm 3.3$ | $74.5 \pm 7.1$ | $59.5 \pm 8.8$ | $79.5 \pm 7.9$ | $17.3 \pm 6.8$ |
| **Echo** | | ✓ | ✓ | ✓ | $\mathbf{99.8 \pm 0.7}$ | $\mathbf{98.8 \pm 1.8}$ | $\mathbf{84.5 \pm 6.3}$ | $\mathbf{76.6 \pm 6.8}$ | $\mathbf{89.3 \pm 5.3}$ | $\mathbf{45.2 \pm 10.9}$ |

Our experimental results are primarily presented through figures and tables, with detailed explanations. We first conduct extensive experiments on MPE, followed by selective validation on SMAC. The MPE ablation studies compare two baseline algorithms: FT-QMIX and FT-VDN. As shown in Figure 16 and Figure 17, FT-VDN demonstrates slightly inferior performance to FT-QMIX across fixed and unfixed delay settings, particularly on REFERENCE. Notably, the RDC-enhanced FT-VDN maintains reasonable delay resistance despite lacking a critic network, confirming the framework's compatibility with non-actor-critic approaches while preserving baseline performance characteristics. Architecture comparisons (Figure 16-19) reveal that GRU-based compensators generally underperform Transformer variants, except on 5m_vs_6m. For brevity, Table 4-9 omit the less competitive FT-VDN and GRU-based results. In SMAC experiments, we employ curriculum learning for the actor to ensure convergence, as discussed previously.

Table 8: Fixed delay test results of SMAC (rewards) with baseline algorithm FT-QMIX.

| | C | DR | H | KD | 0 | 2 | 4 | 6 | 8 |
|---|---|---|---|---|---|---|---|---|---|
| Fixed Delay Value | | | | | 0 | 2 | 4 | 6 | 8 |
| | | | | | 3s_vs_5z | | | | |
| Oracle | | | | | **21.0 ± 0.1** | 21.3 ± 0.4 | 19.0 ± 0.7 | 12.6 ± 0.6 | 9.6 ± 0.6 |
| Base | | | | | 21.9 ± 0.5 | 21.8 ± 0.4 | 21.5 ± 0.4 | 21.6 ± 0.5 | 21.4 ± 0.6 |
| Base | ✓ | | | | 21.6 ± 0.2 | 21.7 ± 0.2 | 21.5 ± 0.2 | 21.4 ± 0.2 | 21.8 ± 0.3 |
| Base | | ✓ | | | 21.7 ± 0.2 | 21.6 ± 0.2 | 22.1 ± 0.3 | 21.9 ± 0.5 | 17.7 ± 1.1 |
| Base | ✓ | ✓ | | | 21.4 ± 0.2 | 21.4 ± 0.2 | 21.5 ± 0.3 | 19.5 ± 0.7 | 13.5 ± 0.7 |
| Flash | ✓ | ✓ | | | 21.4 ± 0.2 | 21.4 ± 0.2 | 21.4 ± 0.3 | 21.4 ± 0.4 | 19.3 ± 0.8 |
| Flash | ✓ | ✓ | ✓ | | 21.2 ± 0.2 | 21.4 ± 0.1 | 21.5 ± 0.2 | 21.8 ± 0.3 | 19.7 ± 0.6 |
| **Flash** | | ✓ | ✓ | ✓ | **21.1 ± 0.1** | **21.1 ± 0.1** | **21.3 ± 0.1** | **21.8 ± 0.3** | **21.8 ± 0.3** |
| Echo | ✓ | ✓ | | | 21.2 ± 0.2 | 21.2 ± 0.2 | 21.3 ± 0.2 | 21.2 ± 0.3 | 17.9 ± 0.8 |
| Echo | ✓ | ✓ | ✓ | | 21.2 ± 0.1 | 21.3 ± 0.1 | 21.4 ± 0.2 | 21.7 ± 0.3 | 20.5 ± 0.4 |
| **Echo** | | ✓ | ✓ | ✓ | **21.0 ± 0.1** | **21.1 ± 0.1** | **21.2 ± 0.1** | **21.3 ± 0.1** | **21.3 ± 0.4** |
| | C | DR | H | KD | 5m_vs_6m | | | | |
| Oracle | | | | | **18.5 ± 0.6** | 4.8 ± 0.1 | 4.0 ± 0.1 | 3.7 ± 0.1 | 3.4 ± 0.1 |
| Base | | | | | 8.5 ± 0.2 | 8.4 ± 0.2 | 8.4 ± 0.3 | 8.3 ± 0.3 | 8.0 ± 0.2 |
| Base | ✓ | | | | 8.4 ± 0.3 | 9.1 ± 0.4 | 9.2 ± 0.4 | 8.9 ± 0.4 | 8.4 ± 0.3 |
| Base | | ✓ | | | 9.3 ± 0.7 | 10.8 ± 0.7 | 10.2 ± 0.6 | 7.5 ± 0.3 | 6.5 ± 0.2 |
| Base | ✓ | ✓ | | | 18.0 ± 0.7 | 15.8 ± 1.1 | 12.0 ± 0.8 | 9.2 ± 0.4 | |
| Flash | ✓ | ✓ | | | 18.3 ± 0.7 | 18.1 ± 0.8 | 17.4 ± 0.8 | 16.1 ± 0.9 | 13.7 ± 0.9 |
| Flash | ✓ | ✓ | ✓ | | 18.5 ± 0.4 | 18.1 ± 0.7 | 17.8 ± 0.7 | 14.4 ± 0.8 | 9.9 ± 0.5 |
| **Flash** | | ✓ | ✓ | ✓ | **18.6 ± 0.5** | **18.5 ± 0.7** | **18.3 ± 0.6** | **16.3 ± 0.7** | **13.9 ± 1.0** |
| Echo | ✓ | ✓ | | | 18.2 ± 0.7 | 17.8 ± 0.7 | 17.5 ± 0.7 | 15.8 ± 1.0 | 14.3 ± 0.9 |
| Echo | ✓ | ✓ | ✓ | | 17.4 ± 0.7 | 16.6 ± 0.8 | 16.3 ± 1.0 | 13.8 ± 0.8 | 11.9 ± 0.8 |
| **Echo** | | ✓ | ✓ | ✓ | **18.6 ± 0.7** | **18.6 ± 0.5** | **18.0 ± 0.8** | **16.5 ± 0.9** | **15.1 ± 0.9** |
| | C | DR | H | KD | 6h_vs_8z | | | | |
| Oracle | | | | | **19.6 ± 0.2** | 11.1 ± 0.3 | 9.2 ± 0.2 | 8.7 ± 0.2 | 8.4 ± 0.2 |
| Base | | | | | 12.7 ± 0.3 | 12.9 ± 0.3 | 12.5 ± 0.2 | 11.9 ± 0.2 | 11.0 ± 0.2 |
| Base | ✓ | | | | 14.0 ± 0.3 | 14.0 ± 0.3 | 13.6 ± 0.3 | 13.2 ± 0.2 | 12.9 ± 0.3 |
| Base | | ✓ | | | 12.6 ± 0.3 | 12.6 ± 0.4 | 11.9 ± 0.3 | 11.0 ± 0.2 | 10.4 ± 0.2 |
| Base | ✓ | ✓ | | | 17.4 ± 0.6 | 18.5 ± 0.5 | 14.5 ± 0.5 | 12.8 ± 0.3 | 11.5 ± 0.3 |
| Flash | ✓ | ✓ | | | 18.9 ± 0.3 | 18.5 ± 0.4 | 16.0 ± 0.8 | 13.3 ± 0.4 | 11.9 ± 0.3 |
| Flash | ✓ | ✓ | ✓ | | 19.3 ± 0.3 | 18.6 ± 0.5 | 16.1 ± 0.5 | 13.6 ± 0.5 | 12.4 ± 0.4 |
| **Flash** | | ✓ | ✓ | ✓ | **19.6 ± 0.2** | **19.1 ± 0.3** | **16.9 ± 0.5** | **14.4 ± 0.5** | **13.0 ± 0.4** |
| Echo | ✓ | ✓ | | | 19.4 ± 0.3 | 19.2 ± 0.4 | 16.8 ± 0.6 | 14.3 ± 0.5 | 13.0 ± 0.4 |
| Echo | ✓ | ✓ | ✓ | | 19.4 ± 0.3 | 18.6 ± 0.5 | 15.1 ± 0.6 | 13.2 ± 0.4 | 12.0 ± 0.3 |
| **Echo** | | ✓ | ✓ | ✓ | **19.7 ± 0.2** | **19.5 ± 0.3** | **17.8 ± 0.5** | **15.3 ± 0.5** | **13.6 ± 0.5** |

Table 9: Unfixed delay test results (rewards) of SMAC with baseline algorithm FT-QMIX.

| | C | DR | H | KD | 0-6 | 3-9 | 0-6 | 3-9 | 0-6 | 3-9 |
|---|---|---|---|---|---|---|---|---|---|---|
| Delay Range | | | | | 0-6 | 3-9 | 0-6 | 3-9 | 0-6 | 3-9 |
| | | | | | 3s_vs_5z | | 5m_vs_6m | | 6h_vs_8z | |
| Oracle | | | | | 21.5 ± 0.4 | 17.2 ± 0.8 | 5.5 ± 0.2 | 3.4 ± 0.1 | 11.9 ± 0.4 | 9.0 ± 0.2 |
| Base | | | | | 21.8 ± 0.4 | 21.4 ± 0.5 | 8.5 ± 0.2 | 8.3 ± 0.3 | 12.9 ± 0.3 | 12.4 ± 0.3 |
| Base | ✓ | | | | 21.5 ± 0.2 | 21.5 ± 0.2 | 9.0 ± 0.3 | 9.0 ± 0.3 | 14.2 ± 0.3 | 13.4 ± 0.2 |
| Base | | ✓ | | | 21.7 ± 0.2 | 22.1 ± 0.4 | 10.7 ± 0.9 | 9.1 ± 0.4 | 12.3 ± 0.3 | 11.4 ± 0.2 |
| Base | ✓ | ✓ | | | 21.5 ± 0.2 | 21.3 ± 0.4 | 17.9 ± 0.8 | 14.6 ± 0.8 | 18.5 ± 0.6 | 13.7 ± 0.5 |
| Flash | ✓ | ✓ | | | 21.4 ± 0.2 | 21.7 ± 0.2 | 17.8 ± 0.7 | 17.2 ± 0.8 | 18.4 ± 0.5 | 14.6 ± 0.6 |
| Flash | ✓ | ✓ | ✓ | | 21.3 ± 0.1 | 21.6 ± 0.2 | 18.4 ± 0.6 | 17.3 ± 0.8 | 18.7 ± 0.4 | 15.3 ± 0.6 |
| **Flash** | | ✓ | ✓ | ✓ | **21.1 ± 0.1** | **21.5 ± 0.2** | **18.4 ± 0.7** | **17.7 ± 0.7** | **19.1 ± 0.3** | **15.9 ± 0.5** |
| Echo | ✓ | ✓ | | | 21.2 ± 0.1 | 21.3 ± 0.2 | 18.0 ± 0.8 | 16.7 ± 0.9 | 19.0 ± 0.4 | 15.8 ± 0.7 |
| Echo | ✓ | ✓ | ✓ | | 21.3 ± 0.1 | 21.6 ± 0.2 | 17.5 ± 0.7 | 16.0 ± 0.9 | 19.0 ± 0.5 | 14.8 ± 0.6 |
| **Echo** | | ✓ | ✓ | ✓ | **21.0 ± 0.0** | **21.2 ± 0.1** | **18.5 ± 0.6** | **17.7 ± 0.7** | **19.5 ± 0.3** | **17.2 ± 0.6** |

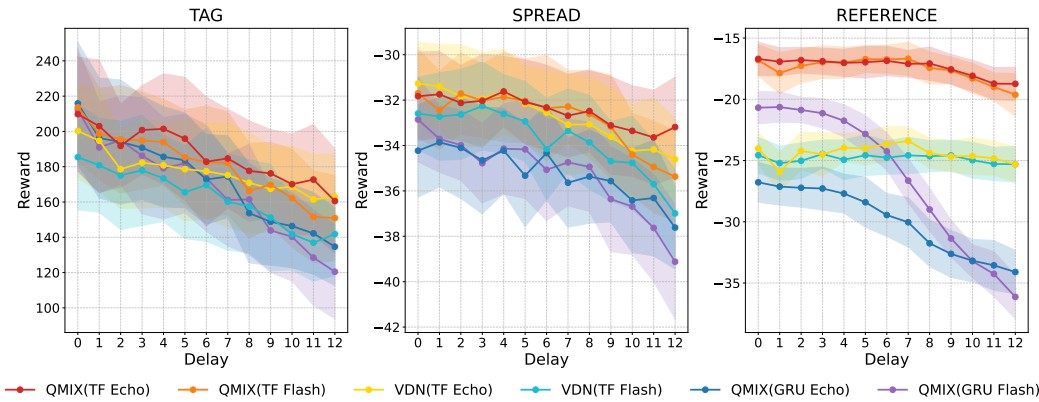

Figure 16: Performance comparison of RDC-enhanced algorithms with different baseline methods and compensator network architectures under fixed delay settings on MPE.

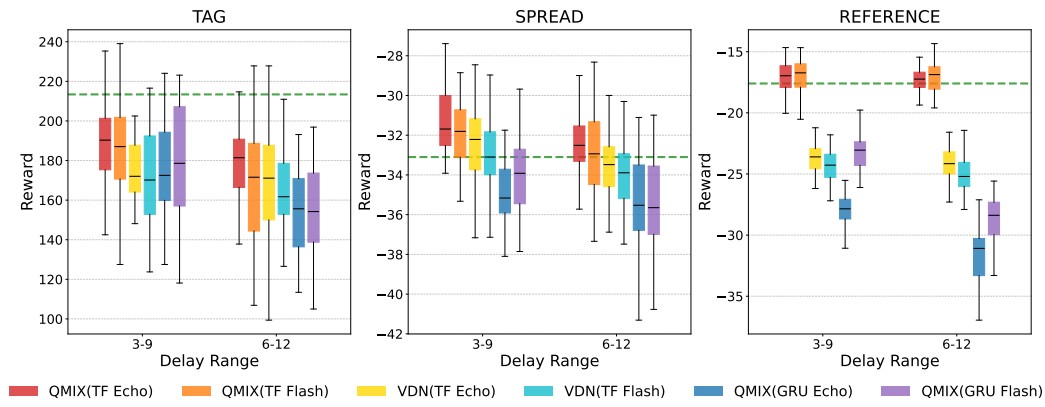

Figure 17: Performance comparison of RDC-enhanced algorithms with different baseline methods and compensator network architectures under unfixed delay settings on MPE.

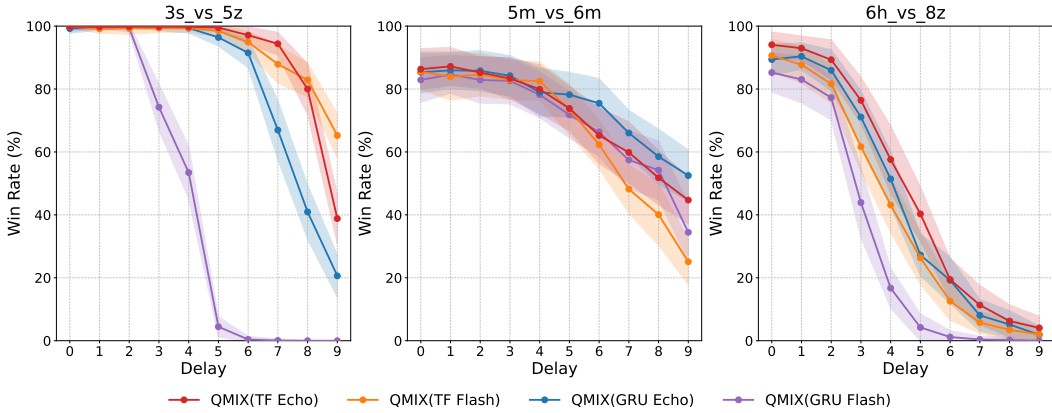

Figure 18: Performance comparison of RDC-enhanced algorithms with different compensator network architectures under fixed delay settings on SMAC.

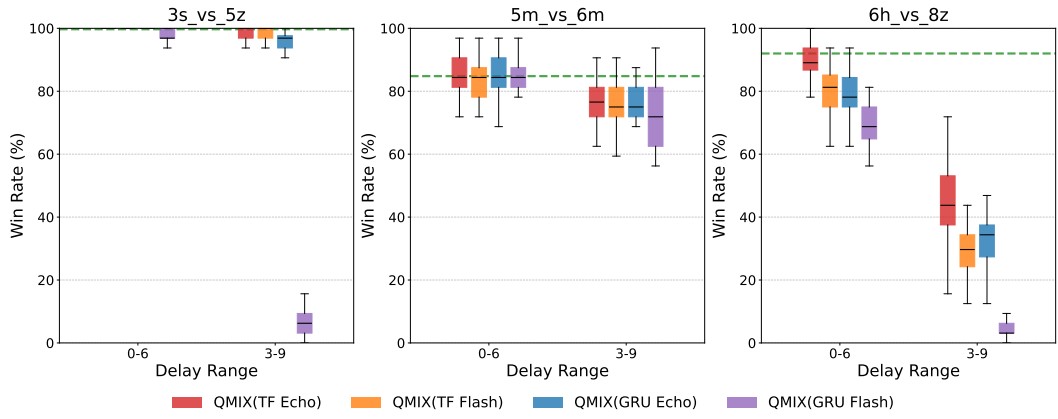

Figure 19: Performance comparison of RDC-enhanced algorithms with different compensator network architectures under unfixed delay settings on SMAC.

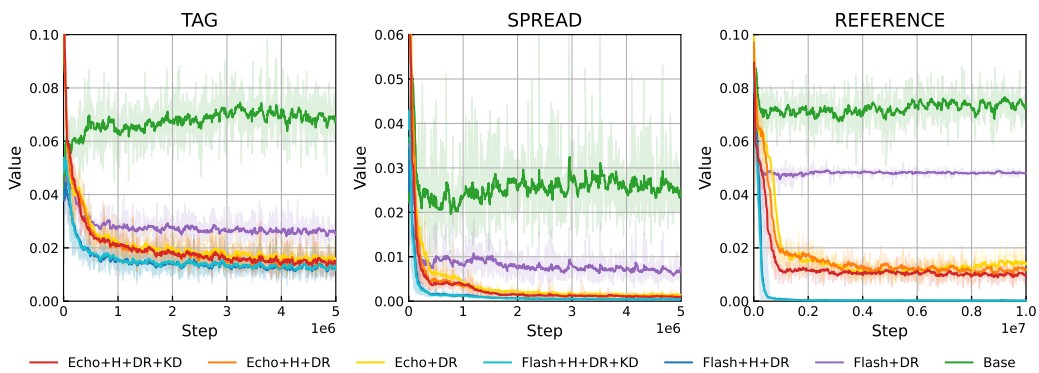

Figure 20: Observation loss curves on MPE with the baseline algorithm FT-QMIX and Transformer compensators.

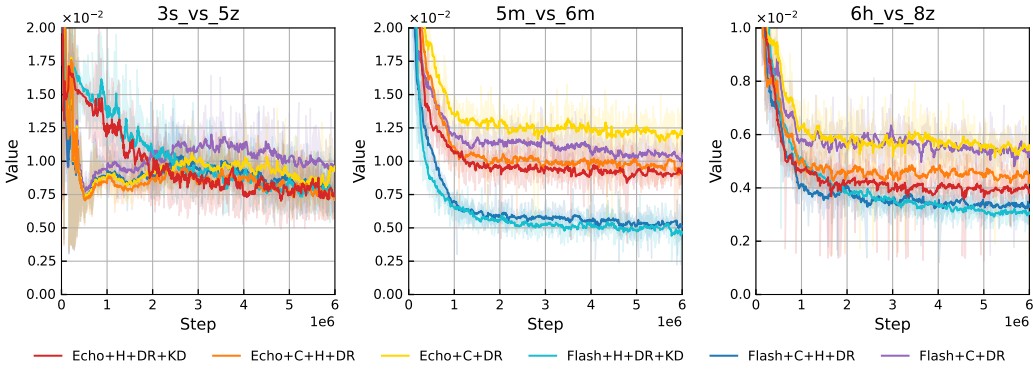

Figure 21: Observation loss curves on SMAC with the baseline algorithm FT-QMIX and Transformer compensators.

Figure 20 and Figure 21 illustrate the variation of compensator loss values with increasing training steps in partial experiments. In SMAC, the observation errors of Base consistently range between 0.1 and 0.2. To distinguish the performance of other algorithms clearly, we omitted the results of Base. The compensator provides substantial improvement in observation accuracy. *Flash* achieves significantly lower compensation errors when utilizing history inputs. On REFERENCE and 5m_vs_6m scenarios, *Echo* exhibits substantially higher compensation errors than *Flash*, a conclusion similarly reflected in non-fixed delay test results. Notably, algorithms employing knowledge distillation achieve superior performance without significant improvement in the compensator, suggesting that addressing delayed observation problems requires simultaneous consideration of both observation compensation and policy formation.

Our experiments are conducted on servers with NVIDIA A30 GPUs (approximately 24GB VRAM). The CPU model is generally unimportant as long as it supports more than eight threads - we use both Intel(R) Xeon(R) Gold 6242R CPU @ 3.10GHz and Intel(R) Xeon(R) Gold 6338 CPU @ 2.00GHz. For the baseline algorithm FT-QMIX with RDC enhancement implemented via Transformer compensator, *Echo* requires an average of 23.8 and 58.3 hours for training on MPE and SMAC, respectively, while *Flash* needs 10.8 and 34.5 hours correspondingly. These time measurements include the teacher model's training duration.

# E   Limitations and Impacts

This paper first theoretically defines DSID-POMDP and proposes the RDC training framework based on the formation process of delayed observation within it. We then discuss the limitations of the proposed method in terms of theory, experimental results, and computational overhead.

- A crucial assumption underlying DSID-POMDP is that each entity in the system only exposes its own state to others. This means that every observation from other entities received by $agent_i$ contains independent and unique information. This assumption fails in environments where agents can relay information through hopping, because when $entity_j$'s information is delayed or missing, $entity_k$ might carry its information. While DSID-POMDP could be extended to accommodate such scenarios, we maintain that this would not represent the optimal approach for defining this class of problems.

- On the experimental front, both Echo and Flash demonstrated poor generalizability on SMAC. As the delay value increased, the win rate declined rapidly. We believe this does not imply that agents inherently cannot achieve victory under such delayed conditions. The current compensator only processes information from the agent's own perspective and does not account for the influence of other agents on the environment. Better compensator designs and training techniques should effectively mitigate this issue.

- The increase in resource overhead with the growing number of agents is a common challenge faced by MARL algorithms. In our experiments, we adopt the practice of having different agents share a single neural network, a setup that is widely used under the CTDE framework. Therefore, although each agent has its own policy network and compensator in terms of structure, the network parameters of all the networks are actually the same during computation. The input to an agent is essentially the observation data, and as the number of agents and entities in the environment increases, the dimensionality of the input will inevitably increase. We believe that this linear increase in dimensionality is acceptable to some extent. When the increase in observation dimensionality significantly impacts the model's inference speed, additional feature extractors may be needed to compress the information. In conclusion, the increase in the number of agents and entities in the environment will not result in a corresponding increase in the number of networks, but rather an increase in the dimensionality of observations.

Despite the limitations above, this paper presents a novel perspective on delayed observation in multi-agent systems and provides a MARL-based solution. Researchers can utilize the RDC training framework in conjunction with baseline algorithms to address application problems across various domains, resulting in a positive societal impact. Our work carries no negative societal implications.

