# OpenReview forum: "Rainbow Delay Compensation: A Multi-Agent Reinforcement Learning Framework for Mitigating Observation Delays"
_NeurIPS.cc/2025/Conference — NeurIPS 2025 poster_

### Official Review · Reviewer_Q4bC · 2025-06-30

**Clarity:** 2
**Significance:** 3
**Originality:** 3
**Rating:** 5
**Confidence:** 4

**Summary:**

This paper tackles a pressing challenge in MARL: handling heterogeneous and stochastic observation delays. The authors introduce both a novel problem formulation and a robust solution framework, making the following core contributions:

1. The authors formalize the setting of stochastic, component-wise observation delays through the Decentralized Stochastic Individual Delay Partially Observable Markov Decision Process (DSID-POMDP). This extends the classical Dec-POMDP framework to realistically capture delay dynamics where different parts of an agent’s observation may arrive at varying times.

2. To address these complex delays, the authors propose RDC, a modular and general-purpose framework comprising four key components:
(a) Compensator: Learns to reconstruct delay-free observations using past delayed observations and actions. Two designs are presented: Flash (one-shot reconstruction) and Echo (autoregressive modeling).
(b) Delay-Reconciled Critic: Incorporates delay-free ground-truth states during centralized training to improve stability and learning quality.
(c) Actor Curriculum: Gradually shifts the actor's input from ground-truth to reconstructed observations to ease the transition and stabilize early training phases.
(d) KD: Leverages a teacher policy trained in a low-delay environment to guide the student in the high-delay setting.

3. Through comprehensive experiments on established MARL environments like MPE and SMAC, under both deterministic and stochastic delay settings, the authors show that traditional MARL methods degrade significantly in delayed settings. In contrast, RDC-based models consistently reduce the impact of delays and, in several cases, approach the performance of an idealized oracle with perfect, delay-free observations.

**Questions:**

1. Following up on W1, what is your primary hypothesis for the compensator's reduced effectiveness on SMAC compared to MPE as delay increases? Does this suggest a need for more advanced sequence prediction models, or perhaps incorporating some form of shared environmental model into the compensation process?

2. In your knowledge distillation experiments, you used a teacher trained on a low-delay setting (delay 1-3) rather than a zero-delay Oracle. You mention that using the Oracle directly as a teacher was unsuccessful. Could you please elaborate on this negative result? What was the failure mode (e.g., training instability, policy divergence)? This is a very interesting practical finding.

3. Have you considered providing the actual delay values $d^{ij}$ as an additional input to the compensator during training, if they were available? Do you think this would simplify the learning problem and improve performance, or is it better for the model to infer delays from the data itself?

**Ethical Concerns:**

["NO or VERY MINOR ethics concerns only"]

**Final Justification:**

I have reviewed the responses from the authors, and will stick to my original score.

**Limitations:**

Please refer to the section on Weaknesses.

**Quality:**

3

**Strengths And Weaknesses:**

Strengths:

1. The paper tackles a problem that is both highly relevant to real-world applications (e.g., robotics, communication networks) and relatively under-explored in the MARL literature. The focus on stochastic, individual component delays is a crucial step beyond prior work that often assumes fixed or uniform delays for the entire observation vector. This finer-grained problem definition is much more realistic.

2. The introduction of the DSID-POMDP is a standout contribution. It provides a clean and formal mathematical foundation for this complex delay problem. The formulation is intuitive and appears correct, serving as a solid theoretical underpinning for the algorithmic solutions that follow.

3. The RDC framework is a well-designed, multi-faceted solution where each component addresses a specific challenge.

4. The paper's motivation is clear, the methods are explained logically, and the figures (especially Figures 2 and 3) are highly effective at illustrating the framework and the core problem. The authors have also provided the source code for reproducibility.

5. Experimental benchmarking is good, and includes necessary ablations and generalizations.


Weaknesses:

1. Generalization Gap on More Complex Tasks: While the results are impressive, there is a noticeable performance gap on the more complex SMAC scenarios compared to MPE. As shown in Figure 9, while RDC significantly improves upon the baselines, the win rate still drops sharply with increasing delay (e.g., in 6h_vs_8z). The paper acknowledges this in Appendix E, but a deeper discussion in the main text about why this might be the case would be beneficial. Is the compensator's task simply too hard with high-dimensional observation/action spaces? Are small prediction errors more catastrophic in strategically brittle scenarios?

2. The DSID-POMDP formalism and the RDC framework implicitly assume that the information from each entity $j$ is unique. As the authors correctly point out in the Limitations (Appendix E), this assumption would be violated in scenarios with information relay (e.g., agent-$k$'s observation contains information about agent-$j$). While extending the model is noted as future work, this is a key assumption that limits the applicability of the current theory to a specific class of MAS.

3. The current approach trains a separate compensator for each agent. While this is standard in MARL, the input to the compensator includes a history of observations from multiple entities (allies and enemies). The paper could be strengthened by a discussion of the computational and sample complexity as the number of agents and entities grows. How does the input dimension and sequence length scale, and what are the potential bottlenecks for applying RDC to systems with dozens or hundreds of agents?

---

> ### Author Rebuttal · Authors · 2025-07-29
>
> &emsp;&emsp;We are honored that you took the time to carefully review our work and provided positive feedback. To better address your questions, we have summarized the weaknesses and questions you pointed out and responded to each of them in detail. We hope this will help clarify any confusion. Thank you once again for your recognition of our work, and we warmly welcome any further questions or suggestions you may have.
>
> **Weakness 1 & Question 1: Following up on W1, what is your primary hypothesis for the compensator's reduced effectiveness on SMAC compared to MPE as delay increases? Does this suggest a need for more advanced sequence prediction models, or perhaps incorporating some form of shared environmental model into the compensation process?**
>
> &emsp;&emsp;You are absolutely right — SMAC tasks indeed have a more complex state space compared to MPE, and under such conditions, even the same level of error can lead to more severe consequences. However, there are many tasks whose complexity far exceeds that of SMAC, which is one of the key motivations behind our proposal of the RDC framework. Rather than aiming for absolute performance superiority, our goal is to provide a structural guideline for future research in this field. The modular design of the RDC framework also allows researchers to easily replace components with more advanced sequence prediction models or world models. When dealing with more complex tasks, such improved models will indeed become essential for effectively handling delayed observations.
>
> **Question 2: In your knowledge distillation experiments, you used a teacher trained on a low-delay setting (delay 1-3) rather than a zero-delay Oracle. You mention that using the Oracle directly as a teacher was unsuccessful. Could you please elaborate on this negative result? What was the failure mode (e.g., training instability, policy divergence)? This is a very interesting practical finding.**
>
> &emsp;&emsp;Thank you for taking the time to carefully read our paper. In our initial plan, we did use Oracle as the teacher model for the delay strategy. However, during actual training, we observed that the model's performance began to decline as knowledge distillation started. In subsequent exploration, we were pleasantly surprised to find that using a low-delay teacher model yielded excellent results. Therefore, we believe that the policy discrepancy is the main reason for the failure of the Oracle teacher model. The experimental results show that the policy differences between the no-delay Oracle strategy and the RDC-enhanced method under delayed observations are significant in the early stages of training. During the knowledge distillation process, the student strategy gradually changes. The larger policy discrepancies may lead to the collapse of the student strategy.
>
> **Question 3: Have you considered providing the actual delay values d^ij as an additional input to the compensator during training, if they were available? Do you think this would simplify the learning problem and improve performance, or is it better for the model to infer delays from the data itself?**
>
> &emsp;&emsp;Your question is very precise. In fact, the actual delay value is part of the compensator's input. We believe this setup can be applied to most multi-agent systems, as delayed information can be accompanied by a timestamp, allowing the receiver to compute the delay value. From a model perspective, this is essential information for completing the prediction. Without the delay value input, we cannot inform the model of the amount of compensation required for different pieces of information. Perhaps in a system with fixed delays, this method could be feasible.
>
> **Weakness 2: The DSID-POMDP formalism and the RDC framework implicitly assume that the information from each entity j is unique. As the authors correctly point out in the Limitations (Appendix E), this assumption would be violated in scenarios with information relay (e.g., agent-k's observation contains information about agent-j). While extending the model is noted as future work, this is a key assumption that limits the applicability of the current theory to a specific class of MAS.**
>
> &emsp;&emsp;We have acknowledged this limitation in our paper. The current theoretical framework does not yet extend to multi-agent systems that involve information forwarding or similar mechanisms. It is foreseeable that, in such scenarios, the observations received by each agent will become more complex. However, this does not make the problem intractable. With appropriate information filtering and simple probabilistic aggregation, it may be possible to extend the theory accordingly. Moreover, the RDC-enhanced method may remain applicable in these cases.
>
> **Weakness 3: The current approach trains a separate compensator for each agent. While this is standard in MARL, the input to the compensator includes a history of observations from multiple entities (allies and enemies). The paper could be strengthened by a discussion of the computational and sample complexity as the number of agents and entities grows. How does the input dimension and sequence length scale, and what are the potential bottlenecks for applying RDC to systems with dozens or hundreds of agents?**
>
> &emsp;&emsp;This is actually a common issue faced by multi-agent reinforcement learning (MARL) algorithms. However, it is essential to clarify that in our experiments, we employed a shared network setup, which is commonly used in the Centralized Training with Decentralized Execution (CTDE) framework for MARL algorithms. Therefore, although each agent has its own compensator in terms of structure, the network parameters of all the compensators are actually the same during computation. The input to the compensator is essentially the observation data, and as the number of agents and entities in the environment increases, the dimensionality of the input will inevitably increase. We believe that this linear increase in dimensionality is acceptable to some extent. When the increase in observation dimensionality significantly impacts the model’s inference speed, additional feature extractors may be needed to compress the information. In conclusion, the increase in the number of agents and entities in the environment will not result in a corresponding increase in the number of compensators, but rather an increase in the dimensionality of observations. If possible, we will add a discussion on this issue in the final submission.

---

> > ### Comment · Reviewer_Q4bC · 2025-08-03
> >
> > Thank you for the clarification. I will keep my rating.

---

> > > ### Author Response · Authors · 2025-08-05
> > >
> > > Thank you for acknowledging the value of our work from the beginning. It truly means a lot to us. The questions you raised have encouraged us to reflect more deeply, and we will ensure that we incorporate these insights into our final submission.

---

### Official Review · Reviewer_vxnb · 2025-07-01

**Clarity:** 2
**Significance:** 3
**Originality:** 3
**Rating:** 4
**Confidence:** 3

**Summary:**

This paper addresses the challenge of observation delays in multi-agent systems. The authors extend the conventional Dec-POMDP framework into a decentralized stochastic individual delay partially observable Markov decision process (DSID-POMDP), which mathematically models multi-agent systems with observation delays. Within this framework, they introduce a technique called rainbow delay compensation (RDC) to handle stochastic individual delays. The proposed method demonstrates notable performance compared to the baseline algorithms under both fixed and unfixed delay settings.

**Questions:**

#### **Maximum delay**

- Is it natural to assume that the existence of a maximum delay? In other words, how is the boundedness of the delay guaranteed?

- In real world, since the distances between agents are not consistent, it seems likely that each agent may experience different maximum delay. Is the maximum delay applied uniformly to all agents?

- In some cases, information may fail to reach the agent (e.g., packet loss). I wonder whether the proposed algorithm can still work properly under such scenarios.

---

#### **Delay distribution**

- In the case of stochastic delays, the performance can vary significantly depending on the delay distribution (e.g., uniform or Poisson). The authors should clarify which delay distribution was used in the experiments and, if possible, briefly evaluate the performance under different distributions.

- The proposed method exclusively focuses on addressing observation delay. However, in real-world environments, various types of delays—such as reward, execution, and inference delays—can occur simultaneously. It remains unclear whether the proposed approach would be effective when these different types of delays occur in combination.

---

####  **Performance of oracle**

- The authors said that the oracle model represents delay-free performance. However, in figures 8 and 9, its performance appears to degrade as the delay increases. This discrepancy should be clarified to avoid unnecessary confusion.

- I am curious how the proposed model outperforms the delay-free performance (oracle). If the proposed method truly performs effective delay compensation, shouldn’t its performance asymptotically converge to the delay-free performance? If the proposed method consistently outperforms the oracle, it may suggest that the improvement stems not purely from effective delay compensation, but rather from other techniques that enhance overall learning performance. It would be great to clarify this point, either through further explanation or additional experiments to support their claim.

**Ethical Concerns:**

["NO or VERY MINOR ethics concerns only"]

**Final Justification:**

I have reviewed the responses, and will raise my score to 4.

**Limitations:**

The proposed method is limited to environments with observation delay only. Therefore, the reviewer cannot guarantee that it will perform equally well in the presence of other types of delays. Moreover, it is also unclear whether the method would maintain its performance in environments where multiple types of delays occur in combination.

**Quality:**

2

**Strengths And Weaknesses:**

#### **Strengths**

- The paper proposes a mathematical framework that extends delay-related problems previously discussed mainly in single-agent systems to multi-agent systems. As real-world applications of reinforcement learning (RL) continue to grow, handling delays in RL has become an increasingly relevant and practical issue. In this context, the paper focuses on an important and timely problem.

#### **Weaknesses**

- In real-world environments, multiple types of delays—such as observation, reward, execution, and inference delays—can occur simultaneously. The proposed method focuses exclusively on observation delay, raising concerns about whether it can still perform effectively in environments where various types of delays occur in combination.

- The proposed approach, termed RDC, feels like an extension of existing techniques originally proposed for single-agent systems [1, 2, 3]. Because of this, the work may lack strong novelty.

---

[1] Firoiu, V., Ju, T., & Tenenbaum, J. (2018). At human speed: Deep reinforcement learning with action delay. arXiv preprint arXiv:1810.07286.

[2] Wu, Q., Zhan, S. S., Wang, Y., Wang, Y., Lin, C. W., Lv, C., … & Huang, C. (2024). Variational delayed policy optimization. Advances in neural information processing systems, 37, 54330-54356.

[3] Wang, W., Han, D., Luo, X., & Li, D. (2023, October). Addressing signal delay in deep reinforcement learning. In The Twelfth International Conference on Learning Representations.

---

> ### Author Rebuttal · Authors · 2025-07-29
>
> &emsp;&emsp;We sincerely appreciate your thoughtful and constructive comments. From your review, we believe you have recognized the value of our work. We have carefully summarized the questions and weaknesses you raised and addressed each of them in detail below. We hope that our responses address your concerns and further highlight the strengths of our study. If possible, we will refine the relevant aspects in the final version based on your feedback. Please don't hesitate to reach out if you have any further questions—we would be happy to address them. All the data provided below follows the same testing settings as described in the main text.
>
> **Question 1.1: Is it natural to assume that the existence of a maximum delay? In other words, how is the boundedness of the delay guaranteed?**
>
> &emsp;&emsp;This is a very reasonable question and one we also considered during our research. We believe that assuming a maximum delay is practical in most real-world applications, as nearly all systems impose an upper bound on tolerable delays. Information exceeding this threshold is typically discarded, which not only ensures bounded delays but also prevents unnecessary resource consumption. As for infinite delays, they can effectively be treated as packet loss.
>
> &emsp;&emsp;You may also be concerned about how the maximum delay is determined. In most practical scenarios, delay fluctuations are relatively stable, allowing users to set a reasonable threshold to avoid excessive data loss. However, in cases where delays are highly variable and hard to estimate, our method may not be suitable.
>
> **Question 1.2: Is the maximum delay applied uniformly to all agents?**
>
> &emsp;&emsp;This is a very practical and relevant question. The RDC framework is specifically designed to handle scenarios where different entities experience different delay distributions, so we do not require a uniform maximum delay across all agents. Instead, it only needs the maximum delay that the system as a whole can tolerate. Importantly, setting this maximum delay does not reduce computational efficiency for agents with shorter delays. In our experiments, we chose, without loss of generality, to use a unified maximum delay across all entities to ensure the clarity and interpretability of the results.
>
> **Question 1.3: Can the proposed algorithm still work in the presence of packet loss?**
>
> &emsp;&emsp;Yes, in such cases, the algorithm still functions as expected. As discussed earlier, information that fails to reach an agent can be treated as having exceeded the maximum delay threshold or as effectively lost. Our experimental environment already includes this mechanism. For example, in step 7, agent A receives data from agent B, which originated from step 5, implying a 2-step delay. In step 8, if the delay exceeds 3, agent A simply uses the most recent available information as the current observation. In other words, the algorithm handles packet loss by consistently using the latest available data. That said, such events are relatively rare during training. In scenarios where packet loss is frequent or complete, the RDC framework would not be applicable, as it is not designed to recover information from absence. Nor is that the intended purpose of our method.
>
> **Question 2.1: Which delay distribution was used in the experiments? How does performance vary under different distributions?**
>
> &emsp;&emsp;Different delay distributions can indeed affect model performance. As stated in Appendix C, we used a uniform delay distribution in our experiments and provided the rationale. We apologize for not emphasizing this point more clearly, which may have led to your misunderstanding. Notably, this highlights a strength of the RDC-enhanced method: it does not require prior knowledge of the delay distribution, nor does it include a learning process to fit specific distributions. The compensator simply restores delayed information regardless of its distribution.
>
> &emsp;&emsp;Theoretically, different delay distributions affect performance by changing the likelihood of certain delay values occurring at different time steps. To further address your concern, we evaluated model performance under several common distributions with delay ranges of 3–9. All models were trained under uniform delay conditions. The results show that performance fluctuations remain within a controllable range. When a distribution's values align well with the delay range the model handles effectively, performance may improve relative to the uniform case. Conversely, if delay values fall outside the model's optimal range, performance may decline slightly.
> ||||||
> |:---:|:---:|:---:|:---:|:---:|
> |Distribution Types|Uniform|Bernoulli(0.5)|Normal(6, 2)|Poisson(6)|
> |TAG-RDC(Echo)|189.2$\pm$22.3|184.1$\pm$32.4|189.8$\pm$29.8|191.3$\pm$24.8|
> |SPREAD-RDC(Echo)|-31.4$\pm$1.9|-31.8$\pm$2.0|-32.0$\pm$1.6|-32.1$\pm$1.8|
> |REFERENCE-RDC(Echo)|-17.0$\pm$1.2|-17.1$\pm$1.3|-17.0$\pm$1.3|-16.8$\pm$1.2|
>
> **Weakness 1 & Question 2.2: Can the proposed method handle combinations of different delays, as seen in real-world scenarios?**
>
> &emsp;&emsp;Thank you for this insightful question. We acknowledge that different types of delays often coexist in real-world systems, but they typically fall under distinct research objectives. In the early stages of our work, we reviewed existing studies on observation delays [1], action delays [2], and reward delays [3]. Prior research [4] has shown that these delay types can be treated equivalently in single-agent settings, and we initially attempted to extend this idea to MARL. However, the results were not encouraging. In multi-agent systems, especially with the observation delays we focus on—more reflective of real-world settings—such equivalence breaks down. Action and inference delays from multiple agents jointly influence environment transitions and are tightly coupled with agents' delayed observations. Without strong independence assumptions, it is not possible to prove that different delay types are equivalent. Therefore, our method specifically targets observation delays in multi-agent settings and does not guarantee high performance when other delay types are present.
>
> **Question 3.1: Why does the oracle’s performance degrade with increasing delay, despite being described as delay-free performance?**
>
> &emsp;&emsp;The Oracle model is trained using a baseline algorithm (e.g., FT-QMIX) in a no-delay environment, and its performance under 0-delay testing reflects the ideal performance that delay compensation methods aim to achieve. While we considered plotting the Oracle's 0-delay performance as a horizontal line to serve as a reference baseline, we found this would not clearly illustrate how performance degrades under delay. We apologize for any confusion this may have caused and hope this explanation clarifies our design choice.
>
> **Question 3.2: Why does the proposed method outperform the delay-free oracle? Shouldn’t it at most converge to the oracle if it only compensates for delays?**
>
> &emsp;&emsp;We provided an explanation for this phenomenon in lines 358–362 of the paper. First, regardless of algorithmic improvements, the performance after delay compensation should theoretically be bounded above by the performance of the no-delay Oracle. In our paper, the observation that the RDC-enhanced method outperforms the ideal Oracle in some scenarios is attributed to the implicit effect of additional training steps. Although the student model undergoes the same amount of training as the Oracle, the extra guidance it receives from the teacher model through knowledge distillation is difficult to quantify. Moreover, another key factor is that the Oracle did not fully converge. In the two scenarios where this phenomenon is most evident—SPREAD and REFERENCE—the Oracle's training curves still show a slight upward trend toward the end. We provide Oracle’s performance with additional training steps in the table below. As shown, the Oracle eventually surpasses the performance of our method with continued training. However, to maintain consistency across all plots in the main text, we have to limit the Oracle's results to the corresponding number of environment steps. We will include a more detailed discussion of this phenomenon in the revised submission.
> ||||||||
> |:---:|:---:|:---:|:---:|:---:|:---:|:---:|
> |Training Steps|5e6|6e6|7e6|8e6|9e6|**1e7**|
> |SPREAD-Oracle|-33.1$\pm$2.0|-32.0$\pm$1.9|-31.1$\pm$1.7|-29.2$\pm$1.6|-29.8$\pm$1.6|**-28.8$\pm$2.0**|
> |Training Steps|1e7|1.1e7|1.2e |1.3e7|1.4e7|**1.5e7**|
> |REFERENCE-Oracle|-17.6$\pm$1.2|-17.1$\pm$1.5|-17.3$\pm$1.4|-17.0$\pm$1.3|-16.6$\pm$1.4|**-16.5$\pm$1.2**|
>
> **Weakness 2: The proposed approach feels like an extension of existing techniques originally proposed for single-agent systems.**
>
> &emsp;&emsp;Our core contribution lies in identifying the previously overlooked mechanism of observation delay in multi-agent systems. Based on this insight, we designed a more accurate and robust compensator architecture. In addition, the use of various curriculum learning strategies also effectively addresses challenges encountered during the training process.
>
> **Reference**
>
> [1] Chen B, Xu M, Liu Z, et al. Delay-aware multi-agent reinforcement learning for cooperative and competitive environments[J]. arXiv preprint arXiv:2005.05441, 2020.
>
> [2] Wang F, Zhang H, Zhang Y. Resolving action delay: Multi-agent reinforcement learning based on state prediction[C]//Chinese Intelligent Systems Conference. Singapore: Springer Nature Singapore, 2024: 552-563.
>
> [3] Zhang Y, Zhang R, Gu Y, et al. Multi-agent reinforcement learning with reward delays[C]//Learning for Dynamics and Control Conference. PMLR, 2023: 692-704.
>
> [4] Wang W, Han D, Luo X, et al. Addressing signal delay in deep reinforcement learning[C]//The Twelfth International Conference on Learning Representations. 2023.

---

> > ### Comment · Reviewer_vxnb · 2025-08-05
> >
> > Dear Authors,
> >
> > Thank you for your thoughtful response. Most of your replies make sense to me. I trust that you will make the necessary modifications in response to my concerns, and I will raise my score to 4.
> >
> > Best regards,

---

> > > ### Author Response · Authors · 2025-08-05
> > >
> > > We sincerely thank you for your valuable suggestions, which have been extremely helpful to us. We will ensure that we provide additional clarification and make revisions concerning the issues raised in the rebuttal in the final version of the paper.

---

### Official Review · Reviewer_Cv3z · 2025-07-02

**Clarity:** 2
**Significance:** 2
**Originality:** 3
**Rating:** 3
**Confidence:** 4

**Summary:**

The authors propose Rainbow Delay Compensation (RDC), a Multi-Agent RL (MARL) framework that aims to address non-fixed, stochastic observation delays in Multi-Agent Systems (MAS). The authors present a modified version of Dec-POMDPS to handle this - decentralized stochastic individual delay partially observable Markov decision process (DSID-POMDP), which adds delay distributions to different agents' observations/entities. They implement these delays in SMAC and MPE, and show that standard value decompositions can struggle with these delays, while their proposed algorithm can better handle these delays.

**Questions:**

1. Why is addressing non-fixed, stochastic observation delay a critical problem in MARL beyond the general statement that "delay issues are ubiquitous"?
2. Could you clarify why the RDC-enhanced methods appear to perform better than the delay-free "Oracle" baseline during training?
3. Based on Figure 10, it appears the performance differences between several of the top RDC variants are not statistically significant. Does this suggest that the most complex configurations of RDC are not always necessary?

**Ethical Concerns:**

["NO or VERY MINOR ethics concerns only"]

**Final Justification:**

I believe this work is interesting and relevant, but the fact that the proposed method is trained for 10m more steps than the baselines feels like an unfair empirical advantage.

I have seen and appreciate the author's new results on training for a similar training time. Unfortunately, I still feel the training steps concern is not fully addressed. Some of the new results help, but also some of it doesn't make sense, e.g. REFERENCE-Base scores move between -28.5 (16 m) and 29.2 (17m) and then -29.6 (similarly with REFERENCE-Base+DR). Although I can accept, maybe it is a mistype, e.g. not including the negative. Furthermore, this is only included on MPE environments, not SMAC, e.g. not in Fig. 7 results (I know doing experiments would take some more training time and might be tough to do over the rebuttal period).

Overall, I do feel this work presents some interesting ideas, but my concerns are not fully addressed, therefore I still lean to a borderline reject.

**Limitations:**

Apart from the mentioned limitations, the complexity of the full RDC method may pose challenges in practical applications requiring careful hyperparameter tuning and implementation, and likely requires significant additional computational resources.

**Paper Formatting Concerns:**

-

**Quality:**

2

**Strengths And Weaknesses:**

## Strengths

Problem:
- Although there is existing work on delays in single-agent RL [1] and MARL with fixed delays [2], this appears to be one of the first works to model non-fixed, stochastic observation delays in MARL.

Method:
- The DSID-POMDP Formalism appears to be a useful contribution, as it provides a way to model non-fixed observation delays in MARL.

## Weaknesses

Problem:
- The motivation for considering observation delays in MARL could be more clearly articulated. From the introduction, it is not specifically clear why this is an important problem to address. The paper could benefit from a more detailed discussion of the challenges posed by observation delays in MARL and why an important problem to address (beyond saying "delay issues are ubiquitous").

Evaluation & Experiments:
- The are known deficiencies in the SMAC v1 environments [1], These environments can be solved by simply observing the timesteps, which means this might not be a useful scenario to test observation delays, as agents have been shown to ignore observations in these environments. I would recommend considering the SMAC v2 environments [1], which have addressed these issues.
- It is unclear why the oracle (no observation delays) baseline performs worse than the proposed method with delays in Figure 7. The explanation in lines 355-357 is not clear. This is unexpected and should be classified better.
- SMAC experiments are typically run for 10 million timesteps, so it is unclear why Figure 7 stops at 6 million.

Method:
- The method proposed has many components (a compensator, a delay-reconciled critic, the curriculum learning for actors, policy distillation), which makes it difficult to understand how each component contributes to the overall performance. Although the paper does provide some ablation studies, it would need to include variants such as the full RDC and no compensator or no distillation to better understand the impact of each component.
- Following up from the previous point, the proposed method and framing was designed for non-fixed observation delays, but from figure 10, it seems the different components do not have a statistically significant impact in performance.

Related Work:
- This work appears closely related to work in robust MARL, e.g. robustness to noise in observations (such as [3]). This field of MARL should be discussed.

References:

1] Wang W, Han D, Luo X, Li D. Addressing signal delay in deep reinforcement learning. In The Twelfth International Conference on Learning Representations 2023 Oct.

2] Chen B, Xu M, Liu Z, Li L, Zhao D. Delay-aware multi-agent reinforcement learning for cooperative and competitive environments. arXiv preprint arXiv:2005.05441. 2020 May 11.

3] He S, Han S, Su S, Han S, Zou S, Miao F. Robust multi-agent reinforcement learning with state uncertainty. arXiv preprint arXiv:2307.16212. 2023 Jul 30.

---

> ### Author Rebuttal · Authors · 2025-07-29
>
> &emsp;&emsp;Thank you very much for your careful reading of our paper and for raising many critical questions. We sincerely appreciate your valuable feedback and are committed to making the necessary revisions in the final submission. After reviewing your comments, we have summarized the identified weaknesses and questions, and we respond to each point in detail below. We genuinely hope that our responses will help resolve your concerns and that you will reconsider the value of our work. Should any further questions arise, we will address them promptly. All the data provided below were obtained using the same testing settings as described in the paper.
>
> **Weakness 1.1 & Question 1: Why is addressing non-fixed, stochastic observation delay a critical problem in MARL?**
>
> &emsp;&emsp;Your question is indeed important. In MARL, the evolving policies of other agents render the learning environment inherently non-stationary [1, 2]. Stochastic observation delays exacerbate this challenge, as agents must not only cope with non-stationarity but also account for the uncertainty introduced by varying delays. These delays also exacerbate the credit assignment problem [1]: misaligned observations and rewards make it more challenging to associate actions with outcomes, thereby hindering effective policy optimization. More fundamentally, stochastic delays violate the Markov assumption by providing agents with inaccurate perceptions of the current state. Moreover, stochastic delays have a greater impact than fixed ones. With fixed delays, agents can adapt and implicitly predict others’ observations over time, forming a kind of cognitive inertia. However, under stochastic delays, such predictions become unreliable, making it harder for actor networks to compensate. We address the importance of handling stochastic delays from four perspectives and will include a dedicated discussion in the Introduction of the final version.
>
> **Weakness 2.2 & Question 2: Could you clarify why the RDC-enhanced methods appear to perform better than the delay-free "Oracle" baseline during training?**
>
> &emsp;&emsp;We apologize for the confusion. This phenomenon is addressed in lines 358–362 of the paper. We believe the RDC-enhanced method outperforms Oracle in the no-delay scenario because it undergoes more training. As noted in line 679, each low-delay teacher model is trained for 1e7 steps before the student model continues training under knowledge distillation for the same number of steps. However, when comparing training curves and test results, we used the same number of steps for both Oracle and RDC student models, which inadvertently favored the latter. In theory, if the Oracle had fully converged, this discrepancy would not arise. However, it is unrealistic to assume convergence within a fixed training budget in practice. To balance performance and training cost, we limited training to what we considered sufficient to support our experimental conclusions.
>
> &emsp;&emsp;As shown in the paper, the RDC-enhanced method outperforms the no-delay Oracle, particularly in the SPREAD and REFERENCE scenarios. In both cases, the Oracle’s training curves were still trending upward at the cutoff point, which supports our statement: “*as evidenced by Oracle's ongoing performance gains at the end of training in both SPREAD and REFERENCE scenarios.*” To further address your concern, we provide Oracle's no-delay performance at different training steps for these scenarios. As expected, the Oracle outperformed our method with fewer training steps. This result further validates our previous conclusion. We will clarify this point more explicitly in the final version to help readers better interpret this seemingly "unnatural" result.
> ||||||||
> |:---:|:---:|:---:|:---:|:---:|:---:|:---:|
> |Training Steps|5e6|6e6|7e6|8e6|9e6|**1e7**|
> |SPREAD-Oracle|-33.1$\pm$2.0|-32.0$\pm$1.9|-31.1$\pm$1.7|-29.2$\pm$1.6|-29.8$\pm$1.6|**-28.8$\pm$2.0**|
> |Training Steps|1e7|1.1e7|1.2e |1.3e7|1.4e7|**1.5e7**|
> |REFERENCE-Oracle|-17.6$\pm$1.2|-17.1$\pm$1.5|-17.3$\pm$1.4|-17.0$\pm$1.3|-16.6$\pm$1.4|**-16.5$\pm$1.2**|
>
> **Weakness 3.2 & Question 3: Does the lack of significant difference in Figure 10 suggest that the most complex RDC configurations may be unnecessary?**
>
> &emsp;&emsp;We appreciate your observation. Modularity is indeed a key principle of the RDC framework. In simpler scenarios, the presence or absence of certain modules may have limited impacts. For example, we did not apply curriculum learning to the actor in MPE tasks. As shown in Figure 10, history inputs have minimal effect in the TAG and REFERENCE scenarios, possibly indicating a weaker reliance on historical information in these cases. However, this does not diminish the value of the RDC framework. The influence of each module depends heavily on the task's underlying dynamics and is often difficult to assess before training is complete. The framework is designed to offer a flexible set of modular features, allowing users to easily enable or disable components through configuration.
>
> **Weakness 2.1: Why choose SMAC for evaluating observation delays instead of the improved SMACv2, which addresses known deficiencies?**
>
> &emsp;&emsp;SMACv2 [3] highlights two main issues with SMAC: (1) its environments are overly deterministic, enabling algorithms to "memorize" rather than "learn"; and (2) it lacks meaningful partial observability. Regarding the first point, we do not believe this undermines the value of using SMAC for MARL research. Notable studies such as QMIX and MAPPO achieved strong results on SMAC, validating it as a useful benchmark for assessing algorithm effectiveness. Moreover, our results suggest that SMAC does not completely ignore observational input—if it did, the yellow curves in Figures 6–9 would have matched ideal performance, which they clearly do not. As for the second point, our work focuses on handling observation delays, not on partial observability. Whether the agent's field of view is too large does not impact our conclusions.
>
> &emsp;&emsp;Additionally, SMACv2 significantly increases computational cost due to its longer training duration, and memory usage scales rapidly with the number of agents. While we acknowledge that SMACv2 is a more challenging benchmark, our aim is not to improve absolute performance but to enhance robustness to delayed observations. SMAC allows us to test this effectively while being more resource-efficient.
>
>
> **Weakness 2.3: SMAC experiments are typically run for 10 million timesteps, so it is unclear why Figure 7 stops at 6 million.**
>
> &emsp;&emsp;In all three SMAC scenarios of our paper, 6e6 timesteps are sufficient for model convergence. This choice helps minimize unnecessary computational overhead, allowing us to allocate more resources to the extensive ablation studies conducted in this work. Below, we report the performance of the Oracle model under varying training durations in the no-delay setting. While sparse sampling introduces some variance, the full convergence curves clearly show that the models converge by 6e6 steps across all scenarios. We will include these convergence curves and the rationale for selecting 6e6 timesteps in the revised version of the paper.
> |||||||||
> |:---:|:---:|:---:|:---:|:---:|:---:|:---:|:---:|
> |Training Steps|**6e6**|7e6|8e6|9e6|1e7|
> |3s_vs_5z-Oracle|**99.7$\pm$1.2**|99.8$\pm$0.8|99.3$\pm$1.3|99.9$\pm$0.5|99.8$\pm$0.8|
> |5m_vs_6m-Oracle|**84.8$\pm$6.7**|84.8$\pm$7.0|85.2$\pm$6.6|84.4$\pm$5.0|83.8$\pm$6.9|
> |6h_vs_8z-Oracle|**92.0$\pm$4.0**|90.4$\pm$5.1|90.2$\pm$6.1|93.4$\pm$3.6|93.0$\pm$4.6|
>
>
> **Weakness 3.1: It needs to include variants such as the full RDC and no compensator or no distillation to better understand the impact of each component.**
>
> &emsp;&emsp;Due to the page limitation, only the most important results are presented in the main text. The complete ablation study results, along with additional analyses, are provided in Appendix D. These include the results for the full RDC, as well as the no compensator and no distillation settings, as you mentioned. We believe our ablation study is already quite thorough, covering nearly all reasonable combinations of component usage.
>
> **Weakness 4.1: This work appears closely related to work in robust MARL, e.g. robustness to noise in observations (such as [4]). This field of MARL should be discussed.**
>
> &emsp;&emsp; You raise a very insightful point. Both our work and studies on robust MARL [4] address the challenge of non-ideal observations. However, they target fundamentally different issues. From a problem perspective, delayed observations concern when information is received—observations are accurate but arrive late. In contrast, robust MARL focuses on what is observed, often dealing with inaccuracies due to sensor noise or adversarial interference. Methodologically, our approach proactively reconstructs no-delay observations from delayed inputs. Robust MARL methods typically adapt passively to observation imperfections without attempting to reconstruct them. In summary, while both lines of research involve imperfect observations, they address distinct and non-overlapping challenges. We will incorporate relevant developments in this area into the Related Works section of our revised submission.
>
> **Reference**
>
> [1] Gronauer S, Diepold K. Multi-agent deep reinforcement learning: a survey[J]. Artificial Intelligence Review, 2022, 55(2): 895-943.
>
> [2] Zhang K, Yang Z, Başar T. Multi-agent reinforcement learning: A selective overview of theories and algorithms[J]. Handbook of reinforcement learning and control, 2021: 321-384.
>
> [3] Ellis B, Cook J, Moalla S, et al. Smacv2: An improved benchmark for cooperative multi-agent reinforcement learning[J]. Advances in Neural Information Processing Systems, 2023, 36: 37567-37593.
>
> [4] He S, Han S, Su S, et al. Robust multi-agent reinforcement learning with state uncertainty[J]. arXiv preprint arXiv:2307.16212, 2023.

---

> > ### Comment · Reviewer_Cv3z · 2025-08-05
> >
> > I thank the reviewer for their detailed response. Most of the replies are sensible.
> >
> > However, for point *"Weakness 2.2 & Question 2: RDC > Oracle "* -- in the rebuttal you mention that each RDC variant receives an additional 10M interaction steps through teacher pre-training before the student’s own 10 M steps, whereas the “Oracle” baseline is plotted after only 10 M steps. This suggests the headline comparison uses unequal training budgets, which could inflate RDC’s apparent advantage. This appears to me like not a fair comparison across all the results.
> >
> > Could you please clarify the total number of training steps per baseline methods and oracle, across the results?

---

> > > ### Author Response · Authors · 2025-08-05
> > >
> > > Thank you for your recognition. I would like to address your question in detail below.
> > >
> > > First, all the results presented in the figures and tables in the paper were obtained under the same training settings. It is imperative to clarify that, since the RDC method involves a knowledge distillation step, an additional 10M training steps are required to pre-train the teacher model before training the final model in every scenario. This has been explicitly mentioned in Sections 4.4 and C.3 of the paper.
> > >
> > > ||||||||
> > > |:---:|:---:|:---:|:---:|:---:|:---:|:---:|
> > > |**Scenarios**|TAG|SPREAD|REFERENCE|3s_vs_5z|5m_vs_6m|6h_vs_8z|
> > > |**Training Steps**|5M|5M|10M|6M|6M|6M|
> > >
> > > There are two main reasons why the training process of the teacher model is not included in the figures of our main text:
> > > (1) In some scenarios, other methods were not trained up to 10M steps, making it difficult to align the curves for a fair comparison.
> > > (2) The teacher model is trained under low-delay conditions, and its performance is therefore not directly comparable to models trained in high-delay environments.
> > >
> > > As mentioned in our rebuttal, Oracle continues to show marginal performance improvements with additional training steps. However, increasing the training steps to 10M or 15M for all scenarios is unnecessary. To more reasonably present the results, we allowed for the RDC method to outperform the no-delay Oracle in some cases slightly and provided explanations for this phenomenon. Our conclusion has always been that RDC can approach the ideal no-delay performance, not that it can surpass it. Please rest assured that we have no intention of exaggerating the effectiveness of RDC or withholding any facts.
> > >
> > > We will include Oracle’s performance with extended training in the final version of the paper to explain this seemingly “unnatural” phenomenon more thoroughly. If you have any further questions, we are fully prepared to respond at any time. Thank you again for your thoughtful feedback.

---

> > > > ### Author Response · Authors · 2025-08-05
> > > >
> > > > As an additional clarification, the table presents the number of training steps for each scenario, covering all baseline methods (including the Oracle), as well as the RDC method (excluding the teacher model).

---

> > > ### Author Response · Authors · 2025-08-07
> > > **A Gentle Reminder for The Discussion Phase**
> > >
> > > Dear Reviewer,
> > >
> > > Thank you for your valuable suggestions on our work. We want to ensure that our responses have adequately addressed your concerns. As the discussion phase is coming to an end in just two days, please let us know if you have any remaining questions. We would be more than happy to respond promptly.
> > >
> > > Best regards.

---

> > > > ### Comment · Reviewer_Cv3z · 2025-08-07
> > > >
> > > > Thanks for the reply.
> > > >
> > > > To clarify, do the other baselines (Base Base+C Base+DR Base+C+DR) also run for 10m less environment steps?

---

> > > > > ### Author Response · Authors · 2025-08-07
> > > > >
> > > > > Yes, all the methods we presented were trained using the same settings. The additional training steps for the RDC method are due to the knowledge distillation process. For example, in the SPREAD scenario, the complete RDC method first trains a low-delay teacher model for 10 M steps. Then, using this teacher model for guidance, the student model is trained for 5 M steps. Other methods, on the other hand, are trained for only 5 M environment steps.

---

> > > > > ### Author Response · Authors · 2025-08-07
> > > > >
> > > > > From another perspective, the training steps for Ours (Echo), Ours (Flash), Base, Base+C, Base+DR, Base+DR+C, Oracle, and the various ablation variants of the RDC method are all the same. It is only when knowledge distillation is enabled that the model is trained under the guidance of a teacher model, which has been trained for 10 million steps, but the total number of training steps remains unchanged. We hope this explanation helps clarify the process for you.

---

> > > > > > ### Comment · Reviewer_Cv3z · 2025-08-07
> > > > > >
> > > > > > Does Ours(Echo), Our etc always knowledge distillation in the results? I am trying to clarify which results exactly use knowledge distillation and which don't.

---

> > > > > > > ### Author Response · Authors · 2025-08-07
> > > > > > >
> > > > > > > Yes, the algorithms labeled as "Ours" in Figures 6 to 9 all use knowledge distillation. In fact, Ours(Echo) in Figures 6 and 8, and Echo+H+DR+KD in Figure 10, refer to the same approach. When comparing with baseline methods, overly long names might cause misunderstandings, so we use "Ours(Echo)" and "Ours(Flash)" to represent the best models achievable under the RDC framework using the Echo and Flash compensators, respectively. In the ablation studies in Figure 10 and Tables 3 to 9, we clearly showcase the specific modules used by each method. In the results you see, only the "Ours" methods and those with the "KD" abbreviation are using knowledge distillation.
> > > > > > >
> > > > > > > Thank you for pointing this out. We will add a description of this detail in the final version to avoid any further misunderstandings.

---

> > > > > > > > ### Comment · Reviewer_Cv3z · 2025-08-07
> > > > > > > >
> > > > > > > > Thanks to the authors for their clarifications.
> > > > > > > >
> > > > > > > > I believe this work is interesting, but I feel the fact that proposed method is trained for 10m steps longer than baselines means it gives an unfair empirical advantage to the proposed method.
> > > > > > > >
> > > > > > > > Although this is mentioned in the Appendix, I feel this should be made more apparent in the main text. The lines 358–362 should mention the 10m extra steps explicitly and to me this text seems to imply that only the oracle is trained for shorter, and not necessarily the other baselines (e.g. Base Base+C Base+DR Base+C+DR). From my perspective, when using Knowledge Distillation (KD), I still believe the number of total environment steps should be kept constant, with these steps distributed evenly between the KD and the post training phase.
> > > > > > > >
> > > > > > > > Due to this reason, I will maintain my score of borderline reject.

---

> > > > > > > > > ### Author Response · Authors · 2025-08-08
> > > > > > > > >
> > > > > > > > > Thank you for your response. We want to clarify that the proposed method does not involve additional training steps for the final model, as only the teacher model undergoes extra training. Therefore, comparing the student model (which experiences the same number of environment steps as the baseline models and the Oracle) is fair. Using the SPREAD scenario as an example, we understand your concern: since the teacher model in the KD-based method is trained for 10M steps and the student model for 5M, one might expect that other methods not using KD should also be trained for 15M steps. We respect this perspective and have conducted additional experiments to address it. Due to space limitations, we have included the corresponding results in the additional comments.
> > > > > > > > >
> > > > > > > > > The training table presents the results of different baseline methods when trained up to the total number of steps used by both the teacher and student models in the RDC method. Please note that the Oracle is trained and tested in a no-delay environment, while all other methods are trained and tested in environments with random delays between 3 and 9 steps. As shown, these methods do not exhibit substantial performance improvements with the extended training, and the proposed method still significantly outperforms the baselines. In addition, we conducted evaluations under fixed-delay settings and observed similar conclusions. To address your concerns, we will adopt your suggestion in the final version of the paper by setting the training steps of all baseline methods equal to the total training steps used in the RDC method, including both teacher and student models.
> > > > > > > > >
> > > > > > > > > Perhaps there is a difference in perspective regarding how to compare our method with the baseline methods fairly. However, the experimental results consistently demonstrate the superiority of the RDC method and the struggles of baseline methods in delayed observation environments under both settings. This suggests that the disagreement has not led to any misunderstanding regarding the overall conclusions. Therefore, we sincerely hope you will reconsider the value of our work. As you mentioned, our research is indeed interesting, and it would be unfortunate if a minor disagreement that does not affect the core findings were to diminish your overall assessment of our work. In any case, we truly appreciate all the thoughtful suggestions you have provided.

---

> > > > > > > > > ### Author Response · Authors · 2025-08-08
> > > > > > > > > **Tables**
> > > > > > > > >
> > > > > > > > > Training Table
> > > > > > > > >
> > > > > > > > > ||||||||
> > > > > > > > > |:---:|:---:|:---:|:---:|:---:|:---:|:---:|
> > > > > > > > > |Training Steps|**5M**|11M|12M|13M|14M|**15M**|
> > > > > > > > > |TAG-Oracle|**213.4$\pm$33.5**|206.3$\pm$31.5|216.1$\pm$29.9|209.5$\pm$32.3|216.1$\pm$31.6|**215.2$\pm$29.7**|
> > > > > > > > > |TAG-Base|**127.0$\pm$18.8**|111.1$\pm$20.8|144.5$\pm$19.3|130.6$\pm$20.9|137.2$\pm$20.4|**140.6$\pm$19.4**|
> > > > > > > > > |TAG-Base+DR|**144.6$\pm$21.1**|142.5$\pm$21.5|158.6$\pm$22.1|142.3$\pm$21.7|147.8$\pm$20.0|**142.4$\pm$21.8**|
> > > > > > > > > |**TAG-Ours(Echo)**|**189.2$\pm$22.3**||||||
> > > > > > > > > |Training Steps|**5M**|11M|12M|13M|14M|**15M**|
> > > > > > > > > |SPREAD-Oracle|**-33.1$\pm$2.0**|-27.4$\pm$2.1|-27.5$\pm$2.0|-26.7$\pm$1.9|-26.4$\pm$1.8|**-26.2$\pm$1.9**|
> > > > > > > > > |SPREAD-Base|**-39.6$\pm$2.0**|-37.1$\pm$2.0|-37.0$\pm$2.1|-36.8$\pm$2.1|-36.4$\pm$2.0|**-35.7$\pm$2.2**|
> > > > > > > > > |SPREAD-Base+DR|**-40.5$\pm$2.2**|-37.5$\pm$2.0|-36.6$\pm$1.9|-35.8$\pm$2.0|-35.8$\pm$2.0|**-36.6$\pm$2.1**|
> > > > > > > > > |**SPREAD-Ours(Echo)**|**-31.4$\pm$1.9**||||||
> > > > > > > > > |Training Steps|**10M**|16M|17M|18M|19M|**20M**|
> > > > > > > > > |REFERENCE-Oracle|**-17.6$\pm$1.2**|-16.6$\pm$1.3|-16.5$\pm$1.2|-16.7$\pm$1.2|-16.4$\pm$1.1|**-16.5$\pm$1.1**|
> > > > > > > > > |REFERENCE-Base|**-29.9$\pm$1.6**|-28.5$\pm$1.4|29.2$\pm$1.3|-29.6$\pm$1.4|-29.4$\pm$1.5|**28.4$\pm$1.6**|
> > > > > > > > > |REFERENCE-Base+DR|**-27.4$\pm$1.6**|-26.1$\pm$1.6|27.0$\pm$1.7|-27.8$\pm$1.4|-27.8$\pm$1.4|**26.3$\pm$1.6**|
> > > > > > > > > |**REFERENCE-Ours(Echo)**|**-17.0$\pm$1.2**||||||
> > > > > > > > >
> > > > > > > > >
> > > > > > > > > Fixed-delay Test Table
> > > > > > > > > |||||||
> > > > > > > > > |:---:|:---:|:---:|:---:|:---:|:---:|
> > > > > > > > > |**Delay Value**|**0**|**3**|**6**|**9**|**12**|
> > > > > > > > > |TAG-Oracle|215.2$\pm$29.7|126.3$\pm$23.3|78.6$\pm$10.3|68.0$\pm$14.9|59.6$\pm$12.5|
> > > > > > > > > |TAG-Base|141.6$\pm$22.4|152.1$\pm$22.9|138.5$\pm$25.0|120.1$\pm$19.2|105.3$\pm$22.4|
> > > > > > > > > |TAG-Base+DR|142.0$\pm$26.4|153.4$\pm$26.6|143.7$\pm$32.0|116.1$\pm$18.3|107.7$\pm$19.5|
> > > > > > > > > |**TAG-Ours(Echo)**|**209.8$\pm$32.3**|**200.8$\pm$25.1**|**182.9$\pm$36.6**|**176.2$\pm$24.8**|**160.5$\pm$29.5**|
> > > > > > > > > |SPREAD-Oracle|-26.2$\pm$1.9|-37.7$\pm$1.3|-51.5$\pm$2.0|-64.4$\pm$2.5|-73.2$\pm$2.4|
> > > > > > > > > |SPREAD-Base|-38.6$\pm$2.3|-35.6$\pm$1.8|-37.4$\pm$2.2|-41.9$\pm$2.5|-45.7$\pm$2.6|
> > > > > > > > > |SPREAD-Base+DR|-37.3$\pm$2.0|-34.1$\pm$1.5|-35.6$\pm$1.9|-40.1$\pm$2.4|-44.4$\pm$2.0|
> > > > > > > > > |**SPREAD-Ours(Echo)**|**-31.8$\pm$1.9**|**-32.0$\pm$2.0**|**-32.3$\pm$1.9**|**-33.1$\pm$2.1**|**-33.2$\pm$2.2**|
> > > > > > > > > |REFERENCE-Oracle|-16.5$\pm$1.1|-22.9$\pm$1.5|-31.2$\pm$1.5|-39.0$\pm$1.3|-44.1$\pm$1.6|
> > > > > > > > > |REFERENCE-Base|-29.4$\pm$1.8|-28.7$\pm$1.4|-30.4$\pm$1.7|-33.9$\pm$1.7|-37.2$\pm$1.5|
> > > > > > > > > |REFERENCE-Base+DR|-29.0$\pm$1.9|-26.0$\pm$1.5|-27.6$\pm$1.3|-31.6$\pm$1.2|-35.3$\pm$1.9|
> > > > > > > > > |**REFERENCE-Ours(Echo)**|**-16.7$\pm$1.4**|**-16.9$\pm$1.1**|**-16.9$\pm$1.2**|**-17.6$\pm$1.2**|**-18.7$\pm$1.4**|

---

### Comment · Area_Chair_9xDv · 2025-08-04
**Please engage with the author rebuttals**

We have two borderline reject reviews for this paper, each of which has been met with a detailed author response. However, there has not been any engagement yet from the borderline reject reviewers. I am calling up these reviewers to engage, indicate if the response was helpful, and seek additional clarifications from the authors if needed. Time is running out to engage with the authors.

---

### Note · Authors · 2025-08-12

We express our gratitude to all the reviewers for their valuable feedback. We are pleased to see the positive evaluations of our work, which mainly include:

1. As RL sees wider real-world use, addressing non-fixed observation delays in multi-agent settings has become a crucial and underexplored challenge. Our work is among the first to tackle it.

2. Focusing on stochastic, per-component delays advances beyond prior work that assumes fixed or uniform delays, offering a more realistic problem definition.

3. The DSID-POMDP provides a clean and formal mathematical foundation for modeling complex delayed observation problems in multi-agent systems, offering a solid theoretical basis for subsequent solutions.

4. The RDC framework is a well-designed, multi-faceted solution where each component addresses a specific challenge.

5. Experimental benchmarking is good, and includes necessary ablations and generalizations.

6. We have provided the source code for reproducibility.

The main concern is how to fairly compare knowledge distillation methods with others. We believe fairness requires the final student model to train on the same number of environment steps as other methods. Reviewer Cv3z suggests counting both teacher and student steps. We ran additional experiments and found that this does not affect our conclusion that our method is superior. We are open to updating results per Cv3z’s suggestion.

We gratefully accept all reviewers’ suggestions, which will improve clarity for readers. We commit to the following revisions in the final submission:

1. Provide more experiments and a detailed explanation for why the RDC-enhanced method outperforms the no-delay Oracle.

2. Add the importance of addressing delayed observations in MARL in the Introduction.

3. Include a robust MARL overview in Related Work.

4. Clarify the challenges under unbounded delays and packet loss in Limitations.

5. Discuss the performance overhead from more agents and higher observation dimensions in the Appendix.

6. Extend training steps for methods without knowledge distillation to match the total steps of the teacher and student models summed.

Within the rebuttal’s length limit, we have addressed the reviewers’ questions thoroughly. We would like to emphasize that while some clarifications or presentation adjustments may be needed, there are no errors in novelty, accuracy, or core findings. We believe this work merits publication and contributes meaningfully to the field.

---

### Decision · Program_Chairs · 2025-09-17

**Decision:**

Accept (poster)

**Comment:**

The authors propose a framework for MARL with observation delays. The reviewers appreciated that this could be a step towards making MARL more applicable in real world scenarios, though also noted that it addresses just one of many possible issues. (I'm don't think that's a fatal criticism if the basic contribution is still worthwhile.)

To address the delay issue, the authors separately train a compensation mechanism, which seems to help. This is good, but there was some concern among at least one reviewer on how the training time for the compensator should be counted. This is not an unreasonable concern because the use of an extra mechanism with its own training regime does change things, and there could be reasonable disagreements about how his training should be reported and counted.

Perhaps the best thing is to aim to be as clear and explicit as possible to avoid misunderstandings. Overall, I am leaning positive on this paper, with the hope that the authors will take the back and forth with reviewer C3vz into account in any revised version to minimize any possibility of confusion/frustration/disappointment in how future readers will interpret the results.